# The inhibitory action of the chaperone BRICHOS against the α-Synuclein secondary nucleation pathway

Dhiman Ghosh [1,7], Felix Torres [1,7], Matthias M. Schneider [2], Dzmitry Ashkinadze [1], Harindranath Kadavath [1,3], Yanick Fleischmann[1], Simon Mergenthal [4], Peter Güntert [1,5], Georg Krainer [2], Ewa A. Andrzejewska [2], Lily Lin[2], Jiapeng Wei[2], Enrico Klotzsch [4]✉, Tuomas Knowles [2,6]✉ & Roland Riek [1]✉

The complex kinetics of disease-related amyloid aggregation of proteins such as α-Synuclein (α-Syn) in Parkinson's disease and Aβ42 in Alzheimer's disease include primary nucleation, amyloid fibril elongation and secondary nucleation. The latter can be a key accelerator of the aggregation process. It has been demonstrated that the chaperone domain BRICHOS can interfere with the secondary nucleation process of Aβ42. Here, we explore the mechanism of secondary nucleation inhibition of the BRICHOS domain of the lung surfactant protein (proSP-C) against α-Syn aggregation and amyloid formation. We determine the 3D NMR structure of an inactive trimer of proSP-C BRICHOS and its active monomer using a designed mutant. Furthermore, the interaction between the proSP-C BRICHOS chaperone and a substrate peptide has been studied. NMR-based interaction studies of proSP-C BRICHOS with α-Syn fibrils show that proSP-C BRICHOS binds to the C-terminal flexible fuzzy coat of the fibrils, which is the secondary nucleation site on the fibrils. Super-resolution fluorescence microscopy demonstrates that proSP-C BRICHOS runs along the fibrillar axis diffusion-dependently sweeping off monomeric α-Syn from the fibrils. The observed mechanism explains how a weakly binding chaperone can inhibit the α-Syn secondary nucleation pathway via avidity where a single proSP-C BRICHOS molecule is sufficient against up to ~7-40 α-Syn molecules embedded within the fibrils.

Synucleinopathies are neurodegenerative diseases associated with the accumulation and aggregation of α-Syn[1–7] that include three major diseases Parkinson's disease (PD), dementia with Lewy bodies and multiple system atrophy (MSA). The disease-relevant role of α-Syn is further supported by mutations in the α-Syn gene (*SNCA*) and the duplication or triplication of *SNCA* that are associated with early-onset PD. α-Syn is an abundant, intrinsically disordered protein (IDP) comprising a positively charged N-terminal domain (residues 1-60), the so-

[1]Institute of Molecular Physical Science (IMPS), ETH Zürich, Vladimir-Prelog-Weg 2, CH-8093 Zürich, Switzerland. [2]Department of Chemistry, University of Cambridge, Lensfield Road, Cambridge CB2 1EW, UK. [3]St. Jude Children's Research Hospital, Memphis, TN, USA. [4]Institute for Biology, Experimental Biophysics / Mechanobiology, Humboldt-Universität zu Berlin, 10115 Berlin, Germany. [5]Institute of Biophysical Chemistry, Center for Biomolecular Magnetic Resonance, Goethe University Frankfurt am Main, 60438 Frankfurt am Main, Germany. [6]Cavendish Laboratory, Department of Physics, University of Cambridge, JJ Thomson Avenue, Cambridge CB3 0HE, UK. [7]These authors contributed equally: Dhiman Ghosh, Felix Torres. ✉e-mail: enrico.klotzsch@hu-berlin.de; tpjk2@cam.ac.uk; roland.riek@phys.chem.ethz.ch

called non-amyloid-ß component (NAC) domain (residues 61-95), and a C-terminal negatively charged segment (residues 96-140)[2]. α-Syn is of particular interest because of its remarkable structural plasticity: (i) α-Syn is an intrinsically disordered protein both in vitro and in cells, which is transiently compacted through the interaction of the positively charged/aromatic N-terminus with its negatively charged/aromatic C-terminus, (ii) α-Synuclein binds membranes in a helical state[8–10], and (iii) is able to form high-density lipoprotein (HDL)-like particles. During pathological aggregation, α-Syn forms different oligomeric species[11] ultimately maturing into a variety of polymorphic amyloid fibrils[12–21]. Studies suggest that amyloid fibril growth of α-Syn occurs via a nucleation-dependent polymerization reaction[22–25]. Following a slow primary nucleus formation, α-Syn fibrils are elongated by the addition of single monomers. In the next step, the amyloid fibrils multiply by fragmentation or can catalyze the formation of new nuclei from monomers on their surfaces−a process known as secondary nucleation that was first described for sickle cell anemia -30 years ago[26]. Fragmentation and secondary nucleation can significantly accelerate the α-Syn aggregation kinetics[23]. Especially the secondary nucleation pathway greatly accelerates the overall aggregation via the absorption of monomeric protein on the surface of amyloid fibrils. It was found that α-Syn monomers interact transiently via their positively charged N-terminus with the negatively charged flexible C-terminal ends of the fibrils[27]. These intermolecular interactions compete with intramolecular contacts in the monomeric form, opening up of the partially collapsed intrinsically disordered states of α-Syn, increase its local concentration, and may align individual monomers on the fibril surface.

Molecular chaperones play a pivotal role in protein folding and protein homeostasis in vivo by transiently binding their substrate, thereby preventing it from aggregating and exerting a loss of function[28–31]. BRICHOS belongs to this class of chaperones. It has been found in a wide range of species like vertebrates, drosophila, nematodes, echinoderms, and lancelets[32,33]. The BRICHOS domain is present in more than 300 proteins, which belong to 12 distinct families. These include Bri1, Bri2, Bri3, GKN1, GKN2, ChM-1, TNMD, pro-SP-C, Arenicin, Group A, and Group C. The different BRICHOS-containing proteins show a conserved pattern constituting a cytosolic part, a hydrophobic domain, a linker region, a BRICHOS domain (-100 amino acid residues long), and a C-terminal region (except for pro-SP-C)[34]. The hydrophobic domain in most BRICHOS-containing proteins is known or predicted to be a single-pass transmembrane (TM) region. Among the aforementioned proteins, Bri2 and proSP-C BRICHOS are of particular interest in the field of amyloid biology as these have been shown to inhibit Aβ42 aggregation both in vivo and in vitro by interfering with secondary nucleation. In addition, Bri2 reduces the Aβ deposition load by acting as an inhibitor of APP processing. Mutations in Bri2 found in familial British and Danish dementias (FBD and FDD, respectively), as well as in Alzheimer's Disease leads to its loss of function, which causes an alteration of the level of APP metabolites[35].

proSP-C BRICHOS has been shown to break the catalytic cycle that generates Aβ42 toxic oligomers during Aβ42 aggregation[36–40] and amyloid formation in vitro. Surfactant protein C (SP-C) is a transmembrane protein present in lung surfactant[41], which is responsible for lowering the surface tension of the alveolar air−water interface[42,43]. SP-C is produced in the alveolar type II cell from an endoplasmic reticulum (ER) integral membrane protein precursor. proSP-C contains four regions: a short N-terminal segment (residues1-23) faces towards the cytosol and is responsible for intracellular trafficking; a transmembrane (TM) part comprising -35 mostly hydrophobic residues (i.e. Ile and Val), the main part of mature SP-C (residues 24−58), a linker region (residues 59−89), and a BRICHOS domain (residues 90−197). The proSP-C BRICHOS domain has been shown to act as a chaperone that targets the SP-C region of proSP-C and prevent its aggregation, thereby assisting its membrane insertion as a TM helix. Like other chaperones, BRICHOS domains have broad substrate specificity. In the case of proSP-C BRICHOS, it interacts with SP-C, Aβ, and medin associated with amyloids in the aortic wall[34].

In order to explore the action of BRICHOS on the secondary nucleation inhibition, we used solution-state NMR to determine high-resolution structures of the inactive trimeric proSP-C BRICHOS domain and of an active monomeric variant obtained via mutagenesis. Furthermore, we show that BRICHOS act as a secondary nucleation inhibitor against both Aβ42 and α-Syn amyloid aggregation and elucidate the binding interaction to both Aβ42 and α-Syn amyloid fibrils.

## Results

### The proSP-C BRICHOS trimeric solution state NMR structure

The proSP-C BRICHOS domain comprising residues 90-197 was expressed as a fusion protein containing a NT* solubility tag[44] separated by a TEV cleavage site in BL21 (DE3*). After expression and cleavage by TEV protease, the BRICHOS domain was obtained and purified via several chromatographic techniques (see methods section). This resulted in the production of pure recombinant proSP-C BRICHOS domain as confirmed by a single protein band at -12 kDa on a BIS-TRICINE GEL in denaturing condition (Supplementary Fig. 1).

Next, the oligomeric state of proSP-C BRICHOS was investigated by a multi-angle light scattering and microfluidic diffusional sizing, which revealed the existence of a stable trimer with the molecular mass of proSP-C BRICHOS of around 34 kDa (one monomeric unit of pro-SP-C BRICHOS is around 11.5 kDa) (Supplementary Fig. 1b). Furthermore, proSP-C BRICHOS was subjected to a microfluidic diffusional sizing platform to determine a hydrodynamic radius of $R_h = 2.93 \pm 0.23$ nm (Supplementary Fig. 1c), which is in agreement with a molecular mass of $39 \pm 4$ kDa assuming a globular fold[45]. This supports again the formation of a homotrimer by the proSP-C BRICHOS domain, which is also in agreement with the X-ray structure (pdb: 2YAD) (Fig. 1a), shown previously.

While X-ray crystallography provided a structure of the proSP-C BRICHOS homotrimer, it lacks about -20% of the amino acid sequence, namely residues Gln151-Phe180 (Fig. 1a). The 3D structure of proSP-C BRICHOS in solution was thus determined following standard NMR procedures (Fig. 1b, Supplementary Table 1, Supplementary Fig. 1d). Figure 1c shows a superposition of the NMR structure represented by 20 conformers with the X-ray structure (pdb: 2YAD) demonstrating a high level of similarity between them. The fold comprises two α-helices that enclose a central five-stranded β-sheet. Following the nomenclature of Wilander et al.,[46] "face A" of the β-sheet packs against helix 1, and "face B" against helix 2, respectively. The two helices are connected by a long, rather well-defined loop comprising Gln151-Phe180, which is missing in the x-ray structure. It covers almost the entire outward-facing side of the β-sheet in the trimer structure via hydrophobic interactions with Met146 and Leu150 located at the beginning of the loop to residues Val141, Tyr122, Tyr113, Met124, Tyr 195, and Ile197 located in the folded core of the protein subunit, as well as Pro178 and Leu181 at the end of the loop to Ile123 in the folded core of the protein subunit. These hydrophobic clusters at both ends of the loop act as position anchors restraining the loop position to cap the most exposed β-strand, Ile194-Ile197.

As shown in the X-ray structure, the large trimer interface features hydrophobic interactions between Phe94, Ala92, Tyr106 of one subunit against Leu101, Val103, Leu188 and Ala184 of the other subunit of the homotrimer plus two salt bridges between Glu135 and Lys114 and Arg139 and Glu191, respectively (Fig. 2a). In addition, the interface includes the polar residues Thr91, Thr93, and Ser95 at the center of the trimer.

There are some solvent-exposed areas of hydrophobic residues that may act as a chaperone active site (Fig. 2b, c). However, considering the large hydrophobic nature of the client with a transmembrane amino acid segment, a much larger hydrophobic solvent-

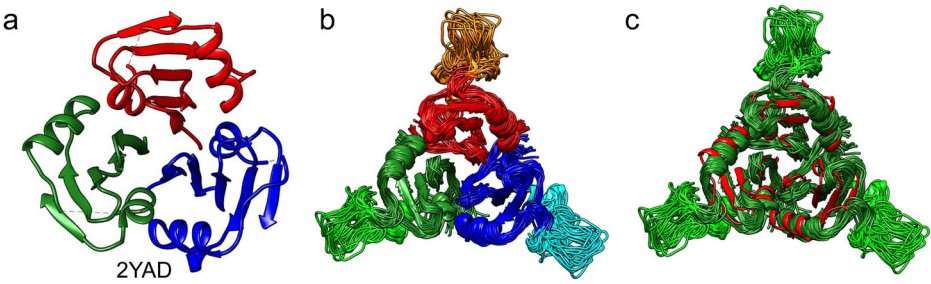

**Fig. 1 | NMR Structure of the proSP-C BRICHOS domain. a** X-ray crystallography structure (2YAD) of the proSP-C BRICHOS domain in a ribbon representation demonstrating the existence of a homotrimer (with the individual entities colored red, green and blue) and **b** the corresponding NMR structure represented by 20 conformers with the long loop forming residues Gln151-Phe180, which are absent in the crystal structure, are highlighted by lighter colors. **c** Superposition of the NMR structure of the proSP-C BRICHOS domain colored in dark green and light green (loop) with the x-ray crystallography structure (2YAD) in red.

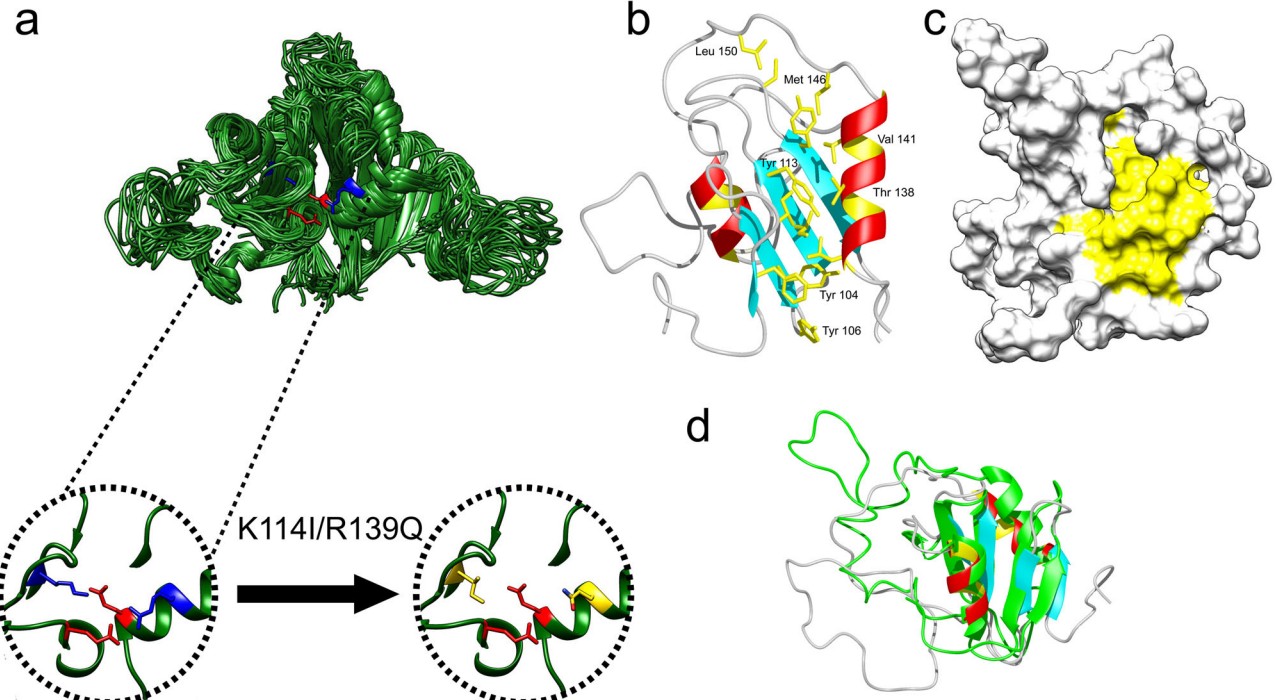

**Fig. 2 | NMR structure of monomeric proSP-C BRICHOS variant II. a** Salt bridge mutation sites of trimeric WT proSP-C BRICHOS that are perturbed in the Var II to generate a stable monomeric form. **b** The solution state NMR structure of the proSP-C BRICHOS mutant (Var II) shows formation of a monomer in contrast to the trimer formed by WT proSP-C BRICHOS. **c** Monomeric proSP-C BRICHOS exhibits an extended hydrophobic cleft as highlighted by yellow color, which is believed to be the active site for the chaperone action. **d** Structure overlay of one monomer from the WT proSP-C BRICHOS trimer (green) and the monomer of the proSP-C BRICHOS mutant (Var II) (cyan for β-sheet) showing that secondary structure elements are conserved in the monomeric proSP-C BRICHOS Var II despite repositioning of the loop.

exposed surface is expected for chaperone activity. It is possible that disassembly of the trimer may cause exposure of several more hydrophobic residues. This suggests that the trimer is an inactive state of proSP-C BRICHOS. This interpretation is in line with chemical cross-linking experiments of proSP-C expressed in transfected A549 cells, which suggested that it does not oligomerize[47]. This has been further supported by peptide binding experiments, which demonstrated that substrate peptide binds to the monomeric proSP-C BRICHOS domain[48] and by the recent findings that monomeric variants of proSP-C BRICHOS domain are more potent inhibitors against Aβ42 fibrillation[49] than its trimeric form.

### The solution state NMR structure of a monomeric variant of proSP-C BRICHOS

Under the hypothesis introduced above that the monomeric from of proSP-C BRICHOS is the active chaperone state, a mutagenesis approach to destabilize the trimer interface was performed. However, the hydrophobic surface/residues of the chaperone might have functional importance. Therefore, in contrast to the approach taken previously[49], where a neutral residue (Ser95) in the interface was replaced by a charged residue (Arg95), we perturbed the salt bridge interaction by mutagenesis at the two interface salt bridges Glu135 - Lys114 and Arg139 - Glu191. These mutations ensure the dissociation of the trimer into monomers while keeping the hydrophobic region intact. The following two mutants were produced by site-directed mutagenesis: proSP-C BRICHOS (E135R, R139E) (denoted below as variant I [Var I]) and proSP-C BRICHOS (K114I, R139Q) (termed variant II, [Var II]) Ile was selected as a replacement, since most of the side chain of Lys114 is within a hydrophobic environment. The degree of oligomerization of the BRICHOS variants was determined by multi-angle light scattering and a microfluidic diffusional sizing assay as described above in the case of WT proSP-C BRICHOS. Multi-angle light

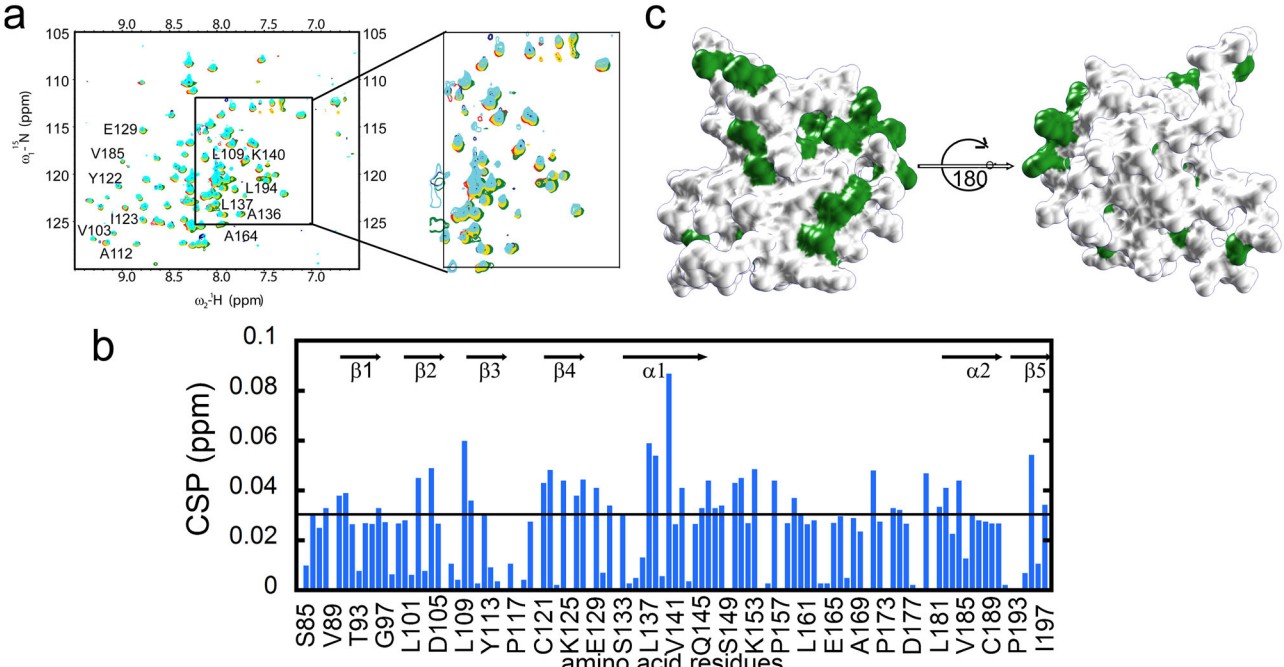

**Fig. 3 | Binding site of a client peptide on the pro-SP-C BRICHOS domain.**
**a** [$^{15}$N,$^{1}$H]-TROSY spectra of $^{15}$N-labeled proSP-C BRICHOS Var II in absence (red) and presence of 200 μM (green), 500 μM (yellow), 1 mM (cyan) and 2 mM (blue) of client peptide B. **b** Client peptide B-induced chemical shift perturbations (CSPs) versus the amino acid sequence of proSP-C BRICHOS Var II with secondary structure elements indicated. **c** Binding sites for interaction of client peptide B mapped on the proSP-C BRICHOS Var II domain. Residues with CSPs > 0.03 ppm are shown in green color.

scattering of variant II indicates a molecular mass of $11 ± 0.55$ kDa ($R_h = 1.81 ± 0.09$ nm) in line with the molecular weight of the monomer (Supplementary Fig. 1c). Consistent with the light scattering data, in the microfluidic diffusional sizing experiment of variant II, the hydrodynamic radius $R_h$ was determined to be $1.65 ± 0.04$ nm (Supplementary Fig. 1c) corresponding to a molecular weight of $9 ± 0.2$ kDa if a globular domain is considered, which is again in agreement with the monomer weight of ~12 kDa. It should be noted that the globular fold is an over simplification[45], as a trimeric, multi-domain protein will have a less compact overall polymer structure, and therefore, a slightly increased hydrodynamic radius is expected.

Next, the 3D NMR structure of variant II in solution was determined following standard procedures for $^{13}$C,$^{15}$N-labelled proteins (Supplementary Fig. 2 and Supplementary Table 1). Variant II was selected for structure determination as it was more stable than variant I. The 3D NMR structure of variant II superimposes well with the trimer structure with an RMSD of 1.26 Å for the core residues (excluding loop) (Figs. 2b, d, Supplementary Fig. 3). The relative orientation of the long loop comprising residues Gln151-Phe180 in relation to the core is however substantially different, extending the surface-exposed hydrophobic cleft. Furthermore, the hydrophobic interface of the trimer is now exposed to the solvent. The long continuous hydrophobic cleft comprises five aromatic side chains with an aromatic triade Phe94, Tyr104 and Tyr106 on one side and the two aromatics Tyr113 and Tyr122 on the other side, connected and extended via Ile96, Leu134, Ile111, Thr138, Val141, Cys120, Leu150, and Met146. The latter three residues are part of the long loop, which is repositioned upon monomer formation. Since hydrophobicity is believed to be key factor for chaperone function, the monomeric structure of variant II is thus considered to represent the chaperone-active form of proSP-C BRICHOS.

### proSP-C BRICHOS - client peptide interaction

In order to study the chaperone active site of proSP-C BRICHOS, the interaction of the monomeric proSP-C BRICHOS Var II with the client peptides KKVVVVVVVKK (peptide A) and VLEMGSGSGSKKVVVVVKK (peptide B) was examined by solution state NMR. VLEMGSG SGSKKVVVVVKK represents part of the transmembrane helical segment of native proSP-C. It may have the capacity to form a β-hairpin and shows a more extended hydrophobic segment. The ligand-induced $^{15}$N and $^{1}$H chemical shift changes of $^{15}$N-labelled SP-C BRI-CHOS Var II were monitored for both peptides A and B (Figs. 3a, b, Supplementary Figs. 4 and 5). In addition, the loss of NMR signal attributed to intermediate exchange binding was also evaluated (Supplementary Fig. 5). While client peptide B binds to proSP-C BRI-CHOS Var II, client peptide A does not (Supplementary Fig. 4). These findings are in line with mass spectrometry (MS)-based results by Wilander et al.[46]. The chemical shift perturbation of client peptide B (Fig. 3a) indicates an interaction on the fast NMR time scale (i.e. faster than ms). The interaction site covers almost the entire hydrophobic cleft as expected (Figs. 3b, c, Supplementary Fig. 5) (note that the probe of the $^{15}$N-$^{1}$H moiety is in the case of the aromatics far away from the interaction site and thus they may not be sensitive to client binding yielding some lack of information in the interaction site). In addition, the charged residue Lys140 appears to be involved in the binding either directly or indirectly. The interactions are considered transient and weak. We calculated the binding affinity between proSP-C BRI-CHOS Var II and peptide B and were found to be ~800 μM (Supplementary Fig. 6). In vivo the interaction between the domain of proSP-C BRICHOS and substrate peptide is likely to be stronger compared to our model peptide (VLEMGSGSGSKKVVVVVKK i.e. peptide B). This could be due to the fact that the substrate peptide is covalently attached to proSP-C in vivo as it is a peptide segment within proSP-C, which further comprises of a hydrophobic stretch more than twice as long as our model system.

It is further noted that more than half of the interaction site lies in the trimer interface of WT proSP-C BRICHOS. Therefore, the monomeric state of the domain is needed for the client peptide interaction as the hydrophobic cleft is easily accessible. This supports the hypothesis that the trimer structure represents an inactive state of

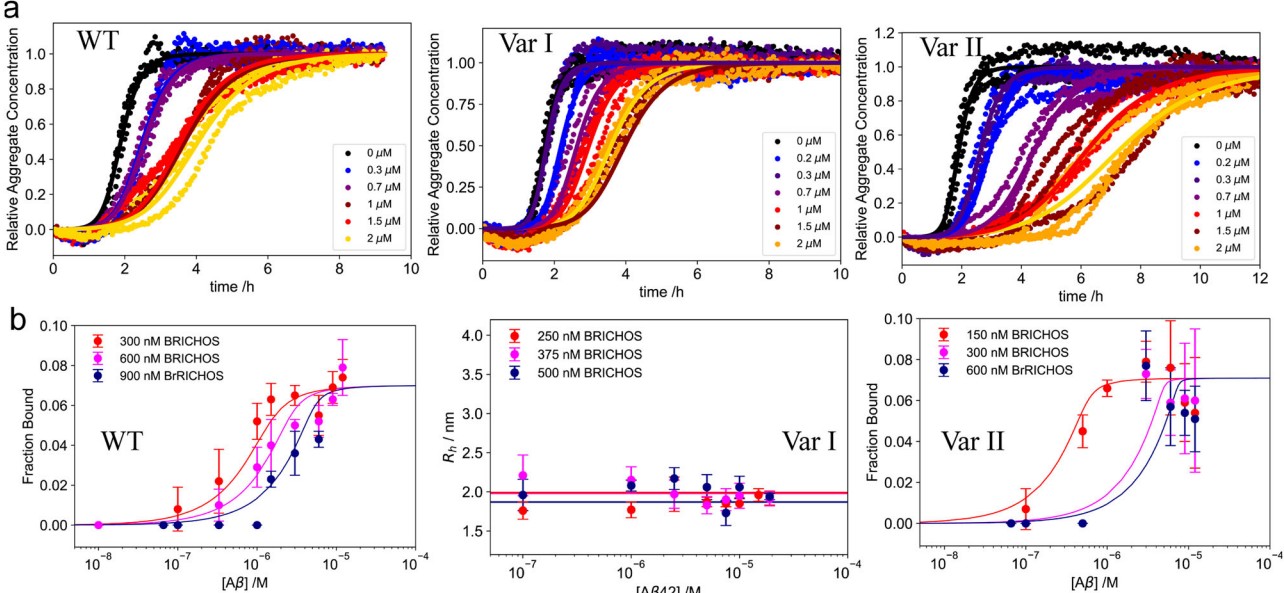

**Fig. 4 | Monomeric proSP-C BRICHOS Var II inhibits Aβ42 fibrillation more efficiently than WT. a** Aggregation of Aβ42 in the presence of different proSP-C BRICHOS variants. Aggregation of Aβ42 in presence of varying molar equivalent of (left panel) WT, Var I (middle panel), and Var II (right panel). The increase in fibrillar mass on the y-axis was measured as an increase in the fluorescence of thioflavin T (ThT). The points represent individual data points. The solid lines represent the fits as obtained from amylo fit as described by Meisl et al. **b** Binding curves for the interaction between different proSP-C BRICHOS variants and Aβ42 fibrils. Left panel: Binding curve for the interaction between Aβ42 fibrils and WT BRICHOS at three different BRICHOS concentrations 300 nM (red), 600 nM (magenta), 900 nM (blue), yielding a dissociation constant, $K_d$ ~ 191.7 [6.2; 517.0] nM with a stoichiometry of 1 BRICHOS molecule per ~8 [5; 12] monomer units in the fibril. Middle panel: Binding curve for the interaction between Aβ42 fibrils and proSP-C BRICHOS variant I at three different BRICHOS concentrations, 250 nM (red), 375 nM (magenta) and 500 nM (blue). The data show no significant binding. Right panel: Binding curve for the interaction between Aβ42 fibrils and proSP-C BRICHOS variant II at three different BRICHOS concentrations, 150 nM (red), 300 nM BRICHOS (magenta), and 600 nM (blue). This data yields a dissociation constant, $K_d$ ~ 21.4 [0.1; 290.4] nM with a stoichiometry of 1 BRICHOS molecule per ~6 [2; 9] monomer units in the fibril. Each experiment was performed 3 times. Error bars are derived from the standard deviation and corresponding mean +/- S.D are represented here.

proSP-C BRICHOS and that the monomer is the active state ready for chaperoning its own transmembrane helix and antiamyloid activity.

## proSP-C BRICHOS inhibits secondary nucleation of Aβ42 during aggregation and amyloid formation

In a first step of kinetic analysis using thioflavin T (ThT) as a fluorescence marker for fibrillar mass[50–56], the impact of WT and the two proSP-C BRICHOS variants on Aβ42 aggregation was investigated. For this purpose, different concentrations of proSP-C BRICHOS variants were incubated with monomeric Aβ42 at pH 8.0 at 37 °C (see Methods). As shown in Fig. 4a, left panel, a clear decrease in the Aβ42 aggregation in the presence of WT proSP-C BRICHOS is observed, suggesting that proSP-C BRICHOS acts as an aggregation inhibitor, as previously reported. The kinetics is best described with a multi-step secondary nucleation model[57], highlighting that proSP-C BRICHOS is a secondary nucleation inhibitor. This is in agreement with previous literature[49]. In contrast to proSP-C BRICHOS Var I, which has lesser impact on the aggregation kinetics (Fig. 4a, middle panel) attributed to an impairment of recognizing the secondary nucleation site of Aβ42, the presence of proSP-C BRICHOS Var II (Fig. 4a, right panel), shows a clear decrease in the secondary nucleation rate. A comparison of inhibitory effect among the different variants of proSP-C BRICHOS against Aβ42 fibrillation kinetics (Fig. 4a) shows that proSP-C BRICHOS Var II has the strongest inhibitory effect compared to the other two variants whereas WT and Var I has almost similar inhibitory effect. To further understand the mechanism of action of proSP-C BRICHOS and its variants in interfering aggregation kinetics, we determined the dissociation constant of different proSP-C BRICHOS variants to Aβ42 fibrils and studied how the binding affinity and stoichiometry correlate with differences in the aggregation kinetics. For this purpose, Aβ42 fibrils of varying concentration were incubated with fluorescently

labeled proSP-C BRICHOS and the two variants. Measuring the hydrodynamic radius as a function of the Aβ42 concentration allows determining both the binding stoichiometry and the dissociation constant ($K_d$)[58]. We found hydrodynamic radius increases with time and saturation is reached after 48 h. Therefore, all samples were subsequently incubated for this time period to ensure equilibrium conditions. The hydrodynamic radius of WT proSP-C BRICHOS was monitored before and after interaction with Aβ42 fibrils. Increase in hydrodynamic radius of WT proSP-C BRICHOS indicates a binding event between WT proSP-C BRICHOS and Aβ42 fibrils. From global fitting, as shown in Fig. 4b, left panel and Supplementary Fig. 7a, the binding affinity was determined to be $K_d = 191.7$ [6.2; 517.0] nM with a stoichiometry of 1 WT proSP-C BRICHOS molecule per ~8 [5; 12] Aβ42 monomer units on the fibril [all the measurements lie within the 95% confidence interval]. This indicates that WT proSP-C BRICHOS is a sub-stoichiometric amyloid fibril inhibitor.

In contrast, for the monomeric proSP-C BRICHOS Var I there was no significant increase in hydrodynamic radius under any conditions used (Fig. 4b middle panel). This indicates that there is no significant binding between proSP-C BRICHOS Var I and Aβ42 amyloid fibrils. (Fig. 4b, middle panel).

Similarly to WT, the interaction between the proSP-C BRICHOS Var II and Aβ42 fibrils showed a significant increase in hydrodynamic radius, demonstrating that proSP-C BRICHOS Var II binds to the Aβ42 fibrils too. From global fitting, as shown in Fig. 4b, right panel, it was possible to determine the binding affinity $K_d = 21.4$ [0.1; 290.4] nM with a stoichiometry of 1 proSP-C BRICHOS Var II molecule per ~6 [2; 9] monomer units in the fibril. As a control, the interaction between the different proSP-C BRICHOS variants to monomeric Aβ42 were measured as well and no statistically significant binding was observed (Supplementary Fig. 8, right panel).

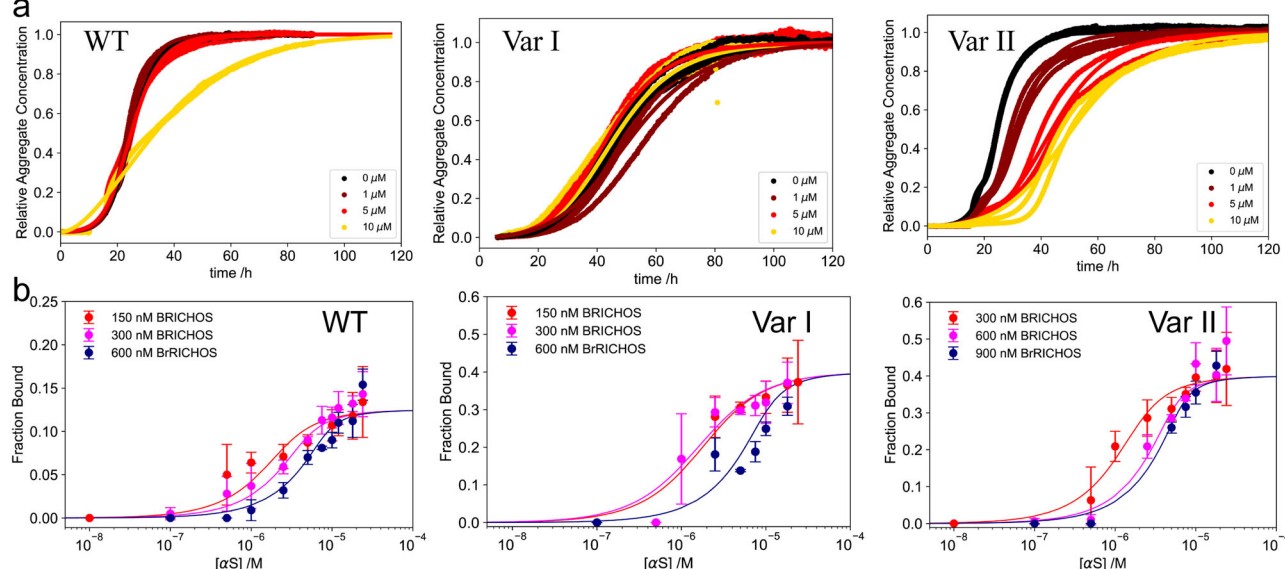

**Fig. 5 | Influence of different proSP-C BRICHOS variants on binding to α-Syn fibrils and α-Syn fibrillation pathway. a** Aggregation of α-Syn in presence of different proSP-C BRICHOS variants. Aggregation of α-Syn in presence of varying molar equivalents of WT (left panel), Var I (middle panel) and Var II (right panel). The increase in fibrillar mass was measured as an increase in the fluorescence of thioavin T (ThT). The points represent individual data points, the solid lines represent the fits as obtained from amylo fit (Meisl et al.). **b** Binding curves for the interaction between different proSP-C BRICHOS variants and α-Syn fibrils. (left panel) Binding curve for the interaction between α-Syn fibrils and WT proSP-C BRICHOS at three different concentrations: 150 nM (red), 300 nM (magenta), 600 nM (blue). This yielded a dissociation constant, $K_d = K_d$ = ~695.7 [263.1, 1311] nM

with a stoichiometry of 1 BRICHOS molecule per ~38 [22; 56] monomer units in the fibrils. (Middle panel) Binding curve for the interaction between α-Syn fibrils and proSP-C BRICHOS Var I at three different concentrations, 150 nM (red), 300 nM (magenta), and 600 nM (blue). This yielded a dissociation constant, $K_d$ ~ 1.26 [0.09; 2.96] nM with a stoichiometry of 1 BRICHOS molecule per ~~18 [12; 60] monomer units in the fibril. (Right panel) Binding curve for the interaction between α-Syn fibrils and proSP-C BRICHOS variant II at three different BRICHOS concentrations: 300 nM (red), 600 nM (magenta), and 900 nM (blue). This yielded a dissociation constant, $K_d$ ~ 450.2 [109.8, 1005] μM with a stoichiometry of -7 [4; 10] monomer units in the fibril. Each experiment was performed 3 times. Error bars are derived from the standard deviation and corresponding mean +/- S.D are represented here.

## proSP-C BRICHOS inhibits secondary nucleation of α-Syn during aggregation and amyloid formation

To explore whether proSP-C BRICHOS can also interfere with other protein aggregation via secondary nucleation inhibition, its effect on the aggregation of α-Syn was studied. As shown in Fig. 5a, all three proSP-C BRICHOS variants have an inhibitory effect on the aggregation kinetics by slowing down secondary nucleation.

Following the kinetic investigation (similar to Aβ42), the binding affinity and stoichiometry between proSP-C BRICHOS variants and α-Syn were studied (Fig. 5b). Unlike the case for Aβ42, the equilibrium with 0.025, 0.125, 0.25, 0.5 molar equivalents of proSP-C BRICHOS and 1 molar equivalent of α-Syn fibrils was reached within 2 hours already, indicating a significantly faster binding equilibrium. We subsequently determined the dissociation constant and binding stoichiometry using microfluidic diffusional sizing. For the WT proSP-C BRICHOS interaction with α-Syn fibrils a $K_d$ = ~695.7 [263.1, 1311] nM was obtained (Fig. 5b, left panel and Supplementary Fig. 7b). Next, stoichiometry was determined to be 1 proSP-C BRICHOS per ~38 [22; 56] monomer units of α-Syn similar to the case of Aβ42. The interaction between BRICHOS Var I and α-Syn (Fig. 5b, middle panel) shows a dissociation constant $K_d$ ~ 1.26 [0.09; 2.96] μM and a stoichiometry of 1 BRICHOS molecule per ~18 [12; 60] monomer units in the fibril. Finally, the binding constant between α-Syn fibrils and variant II was found to be ~ 450.2 [109.8, 1005] nM, with a stoichiometry of 1 BRICHOS molecule per ~7 [4; 10] monomer units in the fibrils. In line with this data, the lack of any saturation transfer from the fibrils to soluble proSP-C BRICHOS Var II indicates a very slow off-rate in line with the nM affinity. It is interesting to note that although WT proSP-C BRICHOS has been defined as an inactive trimeric form of the chaperone, it is able to inhibit α-Syn aggregation and fibrillation, implying disassembly into its active monomeric state. We hypothesized that in the vicinity of α-Syn fibrils some population of the trimers dissociates into monomers caused by a

change of the local microenvironment on the fibril surface. In our previous study we showed that the pH in the vicinity of the α-Syn fibrils is significantly lower (ca 1.5 pH units) compared to the bulk solution[27]. Looking for support of this idea, we thus studied a potential trimer monomer disassembly at acidic pH monitored by multi-angle light scattering. As shown in Supplementary Fig. 9 at a pH 5.8 WT proSP-C BRICHOS partly disassembles to monomer in contrast to pH 6.8. This finding rationalizes the action of trimeric WT proSP-C BRICHOS against α-Syn secondary nucleation fibrillation.

In contrast, no significant size increase was observed for the interaction of both variants and WT proSP-C BRICHOS to monomeric α-Syn (Supplementary Fig. 8b), suggesting that proSP-C BRICHOS does not interact with monomeric α-Syn.

## Mechanism of secondary nucleation inhibition
Towards elucidation of the mechanism of secondary inhibition of α-Syn aggregation by proSP-C BRICHOS, three distinct series of experiments were performed that build upon the previous findings. It has been shown that secondary nucleation of α-Syn occurs through the flexible C-terminal segment of α-Syn fibrils[27]. α-Syn amyloid fibrils contain a C-terminal segment with a fuzzy coat, to which the N-terminal segment of monomeric α-Syn binds transiently. This interaction perturbs the long-range intramolecular interaction, leads to the exposure of the NAC region of monomeric α-Syn. Finally it leads to the folding of the intrinsically disordered monomers to the amyloid formation[27]. In a first series of experiments, it was investigated whether proSP-C BRICHOS Var II strips off monomeric α-synuclein from the fibrils. [15N,1H]-HSQC NMR spectra were measured of 100 μM 15N-labelled monomeric α-Syn, which was added to unlabelled α-synuclein amyloid fibrils at a very high monomer concentration of 540 μM both in absence and presence of proSP-C BRICHOS Var II. With an estimated dissociation constant $K_d$ of ~1 mM[27], about 47% of

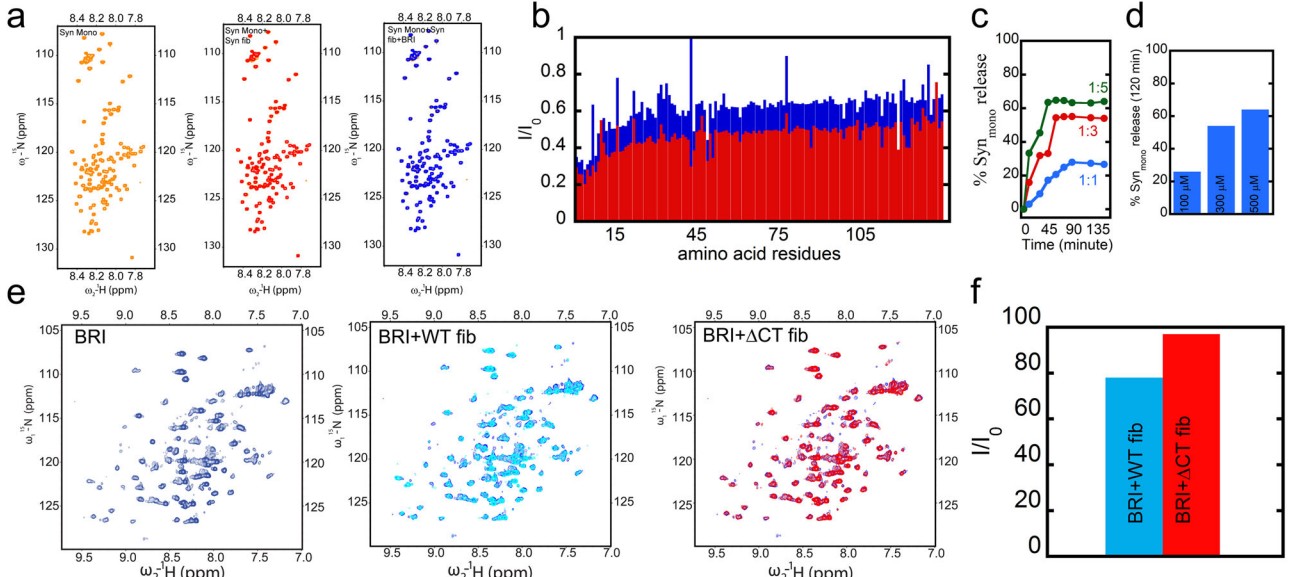

**Fig. 6 | ProSP-C BRICHOS interacts with the flexible C-terminal part of α-Syn fibrils, which is the secondary nucleation site. a** Competition experiment on α-Syn fibrils (Syn fib) with [15N]-labeled α-Syn monomer (Syn Mono) measured by [15N,1H]-HMQC experiments against the addition of proSP-C BRICHOS (BRI) Var II. The [15N,1H]-HMQC of 15N-labeled α-Syn monomer only (orange, left) is the reference spectrum yielding the $I_0$ values for panel b, shown along-side [15N,1H]-HMQC spectra of α-Syn monomer in absence (red, middle) and presence (blue, right) of proSP-C BRICHOS Var II while bound to α-Syn fibrils. **b** Intensity ratios ($I/I_0$) relative to the control measurement with free α-Syn of individual backbone 15N-1H moieties of monomeric α-Syn in presence of α-Syn fibrils (red) or α-Syn fibrils and proSP-C BRICHOS Var II (blue). Signal loss is observed due to the transient binding of monomeric α-Syn to its fibrils (red). Upon addition of proSP-C BRICHOS Var II, signal loss is attenuated, which can be attributed to a competitive binding between proSP-C BRICHOS and monomeric α-Syn on the fibrils. **c** Monomeric α-Syn release attached to α-Syn fibrils upon addition of proSP-C BRICHOS Var II. The

experimental set up is as follows: In a sample with 540 μM unlabeled α-Syn amyloid fibrils 100 μM 15N-labeled monomeric α-Syn is added and incubated for two hours. Next, 100 μM (1:1) or 300 μM (1:3) or 500 μM (1:5) proSP-C BRICHOS Var II were added to the sample at time point 0 and the intensity of the 15N-labeled monomeric α-Syn is measured time-resolved by a 15N-filtered NMR experiment (i.e. [15N,1H]-HMQC) yielding after ca 120 min ~ 25% (for 1:1), ~50% (for 1:3) and ~65% (1:5) monomer bound to fibrils were released as plotted in bar diagram (**d**). **e** [15N,1H]-HMQC spectra of 15N-labeled proSP-C BRICHOS Var II in absence and in presence of WT α-Syn fibrils (blue and cyan, respectively) or in presence of α-Syn(1-121) fibrils (red). Signal attenuation of free proSP-C BRICHOS Var II is observed indicating its binding to the fibrils. **f** Overall intensity ratios ($I/I_0$) of the signals in the [15N, 1H]-HMQC spectra of proSP-C BRICHOS Var II (blue) in presence of WT α-Syn (cyan) or α-Syn(1-121) fibrils (red). $I_0$ corresponds to free proSP-C BRICHOS Var II in buffer, while I correspond to its respective state in presence of the fibrils.

15N-labelled monomeric α-Syn transiently binds to amyloid fibrils, yielding a significant signal decay (Fig. 6a, b). The N-terminal segment of up to ~30 residues shows more signal reduction than the rest of the protein, indicating a direct interaction of the N-terminal 30 residues with the fibrils. This yields also to a signal loss for the C-terminal 110 residues due to enhanced secondary relaxation. Hence, this experiment indicates transient binding of 15N-labelled monomeric α-Syn at approximately every 10th entity within the fibrils. Addition of 100 μM proSP-C BRICHOS Var II to the fibrils containing 100 μM monomeric α-Syn results in an increase of ~25% of the NMR signal (Fig. 6a–d). This modest signal increase indicates that both proSP-C BRICHOS Var II and monomeric α-Syn compete for the same binding site on the amyloid fibrils. The ~25% signal enhancement is approximately in line with the 1.5-fold higher occupancy of BRICHOS on the amyloid fibrils when compared to monomeric α-Syn. In addition, a kinetic experiment was performed that monitored the signal of monomeric α-Syn bound with fibrils upon addition of proSP-C BRICHOS Var II. Over a time period of ~120 min (Fig. 6c, d), a signal enhancement of monomeric α-Syn is observed followed by a plateau, indicating again stripping off transiently interacting monomeric α-Syn from amyloid fibrils by proSP-C BRICHOS Var II. In addition, further signal recovery (up to ~65%) was obtained when proSP-C BRICHOS concentration was increased from ~100 μM to ~500 μM (Fig. 6c, d) as expected for a competition with the same binding site and the knowledge on the higher binding affinities ($K_d$) of proSP-C BRICHOS variants to α-Syn fibrils (i.e. in the order of nM) when compared with the mM binding affinity of α-Syn monomers to its fibrils[27], as well as the kinetic time scale difference between proSP-C BRICHOS Var II that is in the slow time NMR scale (slower than

seconds) while α-Syn monomer binds transiently to its fibrils in the fast NMR time regime (micro-to fast milliseconds).

The hypothesis that both monomeric α-Syn, as well as proSP-C BRICHOS Var II compete for the same binding site on the amyloid fibrils was further investigated. Based on Kumari et al.[27] the secondary nucleation site on the α-Syn amyloid fibrils is supposed to be the C-terminal fuzzy coat. The binding of pro SP-C BRICHOS Var II to amyloid fibrils of WT α-Syn and the C-terminal truncation construct α-Syn (1-121) was therefore studied by NMR (Fig. 6e, f). Indeed, the NMR intensity drop on soluble monomeric 15N-labelled proSP-C BRICHOS Var II in the presence of WT and C-terminally truncated α-Syn amyloid fibrils yielded different results (Fig. 6e, f), where insignificant reduction of NMR signal intensity was observed in case of C-terminal truncated fibrils. Additionally, we monitored the aggregation kinetics of C-terminal truncated α-Syn (1-121) in the presence and absence of proSP-C BRICHOS Var II. It was found that the aggregation kinetics remains unaffected (Supplementary Fig. 10), suggesting proSP-C BRICHOS variants bind to the C-terminal flexible tail of α-Syn fibrils.

Finally, the interplay between proSP-C BRICHOS Var II and α-Syn amyloid fibrils was monitored by time-resolved super-resolution fluorescence microscopy with fluorescently labelled proSP-C BRICHOS Var II in presence of α-Syn amyloid fibrils. For that, fibrils of an average length of around 1 μm were immobilized on a cleaned glass cover-slip and incubated with a Bovine Serum Albumin (BSA)-containing solution to minimize unspecific binding before Atto647 labelled proSP-C BRICHOS was added at different concentrations. To visualize single proSP-C BRICHOS Var II trajectories, total internal reflection fluorescence (TIRF) microscopy was applied. As shown in Fig. 7a and Supplementary

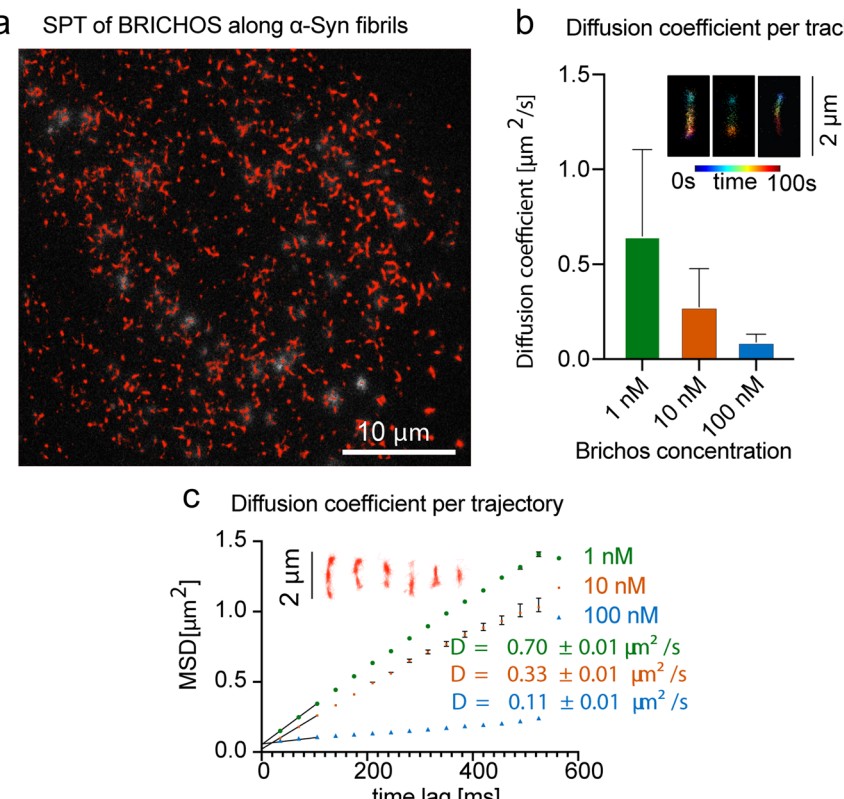

**Fig. 7 | Single particle tracking microscopy of ProSP-C BRICHOS Var II shows 1D diffusion along α-Syn fibrils. a** Single tracks of proSP-C BRICHOS Var II at 10 nM are overlaid with a TIRF microscopy frame showing all trajectories that are longer than 10 frames. **b** Diffusion coefficients are calculated for at least 1000 trajectories and filtered to exclude static molecules ($D > 0.05$ μm²/s). Error bars represent standard deviation. The insets depict 3 examples of long trajectories (color code represents a time of 100 seconds). **c** The mean square displacement for all trajectories is plotted versus the time lag. The first 3 data points are taken to estimate the slope. At higher concentrations (10 and 100 nM) the slope at high time lags, suggesting a sub diffusional character. The insets are six representative isolated tracks along α-Syn fibrils from **a**. For 1nM, 10nM and 100nM no of data points are collected 8545, 2445 and 757, respectively. Error bars are derived from the standard deviation and corresponding mean +/- S.D are represented here.

movie 1, 1D diffusion along fibrils is observed. In order to be able to observe single resolvable trajectories, the concentration was chosen in the nM range. Interestingly, at low concentrations of 1 nM, proSP-C BRICHOS Var II diffuses with 0.7 μm²/s, while higher concentrations resulted in slowing down to 0.1 μm²/s (Fig. 7b). This suggests that the proSP-C BRICHOS Var II density of bound molecules on the fibrils leads to an occupation of available binding sites, thus slowing down the observed 1D diffusion. While single-particle tracking resulted in a mean track length of 12 frames and approx. 350–400 ms residence time on the fibrils (Fig. 7c), the inset in Fig. 7b shows three exemplary time color-coded molecules with residence times of around 100 seconds. This time-resolved observation can explain how one proSP-C BRICHOS Var II molecule is able to interfere with the secondary nucleation process against -7 α-Syn because proSP-C BRICHOS Var II relocates with a short residence time along the fibril axis covering several secondary nucleation sites within a short time frame. However, another possibility of slowing down of 1D phenomenon could be due to the aggregation of BRICHOS molecules on the α-Syn fibrils. To exclude this probability, we performed another set of Single Particle tracking analysis (Supplementary Fig. 11). We compared the diffusion coefficients at the onset of the experiments and after 15 min of exposing α-Syn fibrils with 100 nM fluorescently-labeled proSP-C BRICHOS Var II. However, we did not observe a significant difference in the diffusion coefficients. This suggests availability of binding sites is not drastically changed throughout the experiments. We furthermore plotted integrated proSP-C BRICHOS Var II locations along fibrils, representing all detected locations throughout the time course of the experiment. The obvious higher abundance of bound proSP-C BRICHOS Var II along

α- Syn fibrils restricts the available binding sites. This results in sub diffusive character, which shows non linear behaviour against concentration as expected would be from a 1-D diffusional model. To exclude that we have a dominant fraction of aggregates of proSP-C BRICHOS Var II on the fibrils, we plotted the photons per single molecule detection as a histogram showing a dominant fraction of monomeric proSP-C BRICHOS Var II. However, as the nature of this experiment also contains photobleaching, we cannot fully exclude that already photobleached proSP-C BRICHOS Var II molecules block a subset of the available binding sites. A maximum concentration of 100 nM proSP-C BRICHOS Var II was chosen to exclude overlap of single particle tracks overlapping. The photon statistics can be found in Supplementary Fig. 11 for the different concentrations 1 nM–100 nM, showing no significant differences in the distribution of single molecule brightness. This indicates that no collisions or aggregations occurred.

## Discussion

Molecular chaperons are involved in protein misfolding diseases, including amyloid-related disorders[59–61]. Improper chaperone activity has been shown to cause amyloid diseases in model in vivo systems and could be a risk factor for disease including Parkinson's disease[62,63]. In return, enhanced chaperone presence can interfere effectively with protein aggregation both in vitro as well as in vivo[64–69]. Chaperones may thereby act at various steps during the amyloid aggregation kinetics, including transient interaction with the monomer species as shown for α-synuclein with HSP70/HSP90 chaperones, primary nucleation, fibril elongation, and secondary nucleation.

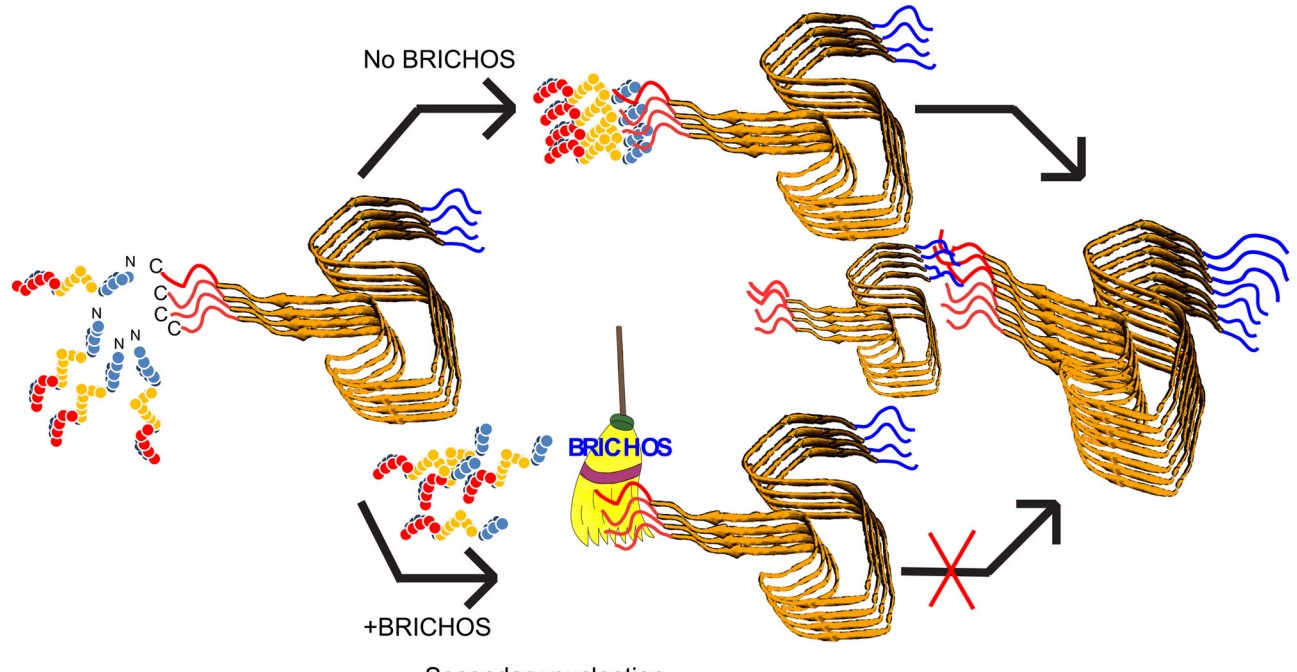

**Fig. 8 | Schematic representation of the secondary nucleation inhibition of α-Syn aggregation and fibrillation by the chaperone proSP-C BRICHOS.** α-Syn monomers converted into fibrils during its aggregation pathway. Monomeric synuclein binds to the C-terminal flexible tail of the fibrils allowing acceleration of the aggregation process via secondary nucleation. ProSP-C BRICHOS competes with the same binding site on the fibrils, preventing further binding of monomers and cleaning the existing monomers that are already bound to the fibrils.

The mechanism of secondary nucleation inhibition against α-Syn fibrillation was investigated on the example of the chaperone proSP-C BRICHOS. For this purpose, the 3D solution state NMR structures of the chaperone inactive trimer and active monomer were determined (Figs. 1 and 2). In contrast to the trimeric form, the monomer contains a large hydrophobic cleft on the surface that serves as a binding site to clients (Fig. 3). The interactions of a hydrophobic peptide flanked by charged amino acid residues appears to be of transient nature and rather weak (i.e. in the high μM to mM range). The weak chaperone action of the BRICHOS domain with the peptides are however interesting within the context of amyloid fibril interaction since it interacts with the flexible C-terminal part of α-Syn comprising both negative charges and some hydrophobic side chains (Ala, Ile, Val, Leu, Try, Met, Ser, Gln, Glu, Phe, Asp), forming the fuzzy coat of the amyloid fibrils. However due to the repetitive structure of amyloids having every 4.7 Å another C-terminal tail, avidity is present yielding binding affinities in the range of hundreds of nM at a stoichiometry of 1 BRICHOS domain to ~7-40 α-synuclein molecules within the amyloid (Fig. 4 and Fig. 5). The BRICHOS domain thereby competes with monomeric α-Syn on the fuzzy coat of amyloid fibrils via transient interaction at a stoichiometry of roughly 1:10, building the basis for secondary nucleation reduction. Based on the single particle tracking and super-resolution microscopy data shown in Fig. 7 the BRICHOS domain moves diffusion-limited along the fibrillar axis sweeping off monomeric α-Syn from the fibrils as demonstrated by time-resolved NMR studies (Fig. 6 and Fig. 8). The sweeping is likely to be possible only if the binding of the proSP-C BRICHOS domain to a single C-terminal segment is weak and transient. However, further evidences are needed to fully establish the fact that the attachment process controls the secondary nucleation mechanism.

To strengthen the concept that sweeping of BRICHOS is relevant for secondary nucleation interference, a kinetic model with available quantitative and approximated information was established (see Supplementary Information and Supplementary Fig. 12). The calculations demonstrate that proSP-C BRICHOS interferes with secondary nucleus formation on the surface of the fibrils by diffusing through thousands of potential nucleation sites and kicking thereby α-Syn monomers off the nucleation site via competition. It is the combination of competitive interference with α-Syn monomers and visiting many nucleation sites by diffusion faster than secondary nucleus formation that builds the mechanistic base of its secondary nucleation interference.

In summary, the proposed mechanism of chaperone action of proSP-C BRICHOS and its variants interfering with secondary nucleation is as follows: Monomeric α-Syn in solution is an intrinsically disordered protein having transient hydrophobic and charge-charge intramolecular interactions between the positively charged N-terminus and negatively charged residues along with hydrophobic residues at the C-terminus[70] leading to the interference of α-Syn folding and aggregation[13]. The monomeric α-Syn interacts transiently in the micro to mili-second range through its N-terminus with the C-terminal segment on the fibrils. This intermolecular interaction interferes with the corresponding intramolecular interaction and thus unfolds further the intrinsically disordered state of α-Syn, aligning the entities and concentrating them on the fibril surface resulting in amyloid surface-catalyzed aggregation. The chaperone proSP-C BRICHOS and variants thereof compete with the synuclein monomer at the same binding site in the C-terminal segment of α-Syn fibrils, thereby sweeping the monomers along the fibrillar surface and cleaning away weakly bound α-Syn during the onset of amyloid formation.

Since secondary nucleation can be the prominent mechanism of amyloid formation, it can be envisioned to interfere with the disease by designing secondary nucleation inhibitors. Indeed, the use of proSP-C BRICHOS domains has been proposed for the treatment of Alzheimer's disease[71]. Therefore, our current study and proposed mechanism may be beneficial for designing other molecules that mimic the activity of the proSP-C BRICHOS as a

chaperone. This may include antibodies or small molecules that bind to the C-terminus part of α-Syn of amyloid fibrils. Based on the mechanism it is indicated that binding affinities of such molecules should not be very high (i.e. rather in the high nM range than in the low nM range) as sweeping is important.

In conclusion an inhibitory mechanism against the secondary nucleation pathway of the Parkinson's disease-related protein α-Syn by the chaperone proSP-C BRICHOS has been elucidated. It can be summarized as competitive sweeping along the secondary nucleation sites on the amyloid fibrils like a brush cleaning a surface.

## Methods

### Recombinant protein expression and purification of α-Syn

N-terminally acetylated, human WT α-Syn was produced by co-expressing the pRK172 plasmid with the yeast N-acetyltransferase complex B (NatB)[72] in BL21(DE3*). α-Syn was expressed and purified according to the previously published protocol with slight modification[25]. Briefly, a single colony from the overnight trans-formed plate was picked up. It was grown in 100 ml LB media containing 100 µg/ml Ampicillin at 37 °. 25 ml of the overnight growth culture was added to freshly prepare 1 lit LB media containing 100 µg/ml Ampicillin and grown further until O.D reached 0.8. IPTG (inducer) was added at a final concentration of 1 mM in the media to start protein expression. The cells were grown at 37 ° for more 4 hours and harvested. α-Syn was obtained from periplasmic extract using nondenaturing protocol. Finally pure α-Syn was obtained by two more round of chromatography purification, ion exchange followed by FPLC attached with a hydrophobic column. It was extensively dialyzed against water, lyophilized and stored at -20 ° until further use. For labeled protein expression, minimal media containing $^{15}$N ammonium chloride was used. Double antibiotic selection i.e. Ampicillin and chloramphenicol was used throughout the expression procedure.

### Recombinant protein expression and purification of Aβ42

Aβ42 comprising residues 1–42 was expressed and purified as previously reported[36]. In brief Aβ42 was purified on a Superdex 75 increase 10/300 g/L column (GE Healthcare, US) on an AKTA purification system (GE Healthcare, US) with a flow rate of 0.7 mL/min and sodium-phosphate buffer (pH 8.0) as elution buffer.

### Recombinant protein expression and purification of proSP-C BRICHOS and variants I and II thereof

The plasmid containing the proSP-C BRICHOS domain sequence, along with a His tag at the N-terminus, NT* solubility tag[73] followed by TEV sequence in a pET- 30a (+) vector, was obtained commercially from GenScript. The transformation was performed in the BL21(DE3*) pLysS cell line in presence of kanamycin and Chloramphenical antibiotics. The cells were grown at 30 °C in M9 media where $^{15}$N ammonium chloride and $^{13}$C glucose were used for spin half Nitrogen and carbon source, respectively. When the O.D reaches 1.2, IPTG was added at a final concentration of 500 mM and the cells were further grown for 16 hours at 30 °C. Finally, the cells were harvested and resuspended in lysis buffer containing 50 mM TRIS, 150 mM NaCl, 5 M Urea, pH ~8.0. The solute was passed through a microfluidizer three times for cell lysis. The fusion protein was then passed through a Ni NTA-column as a first round of protein purification followed by dialysis against 50 mM TRIS - 100 mM NaCl, pH ~8.0 for 24 hours. This ensures proper refolding of the fusion protein. TEV digestion was performed at a 10:1 molar ratio (Fusion protein: TEV) at 4 °C for 8 hours. The pure BRICHOS domain was collected by performing another round of Ni-chromatography and followed by size exclusion chromatography against NMR buffer (25 mM phosphate, 25 mM NaCl, 0.01% NaN$_3$ pH ~6.8).

### Generation of α-Syn and Aβ42 fibrils

α-Syn fibrils were generated in two generations. First-generation fibrils were obtained by incubating monomeric α-Syn in PBS, pH 7.4, typically at concentrations of 300 µM, at 50 rpm and 37 °C for 72 h. The fibrils were centrifuged for 30 minutes at 21000 x g (Centrifuge 5425 R, Eppendorf, Hamburg, Germany), the supernatant was removed, and resuspended in pure PBS. This was repeated twice to wash any residual monomer. Then, the pure fibrils were resuspended in PBS (pH 7.4) and sonicated with a probe sonicator (Bandeline, Berlin, Germany, settings: power: 10%, cycle: 30%, duration 30 seconds). For generation 2 fibrils, generation 1 fibrils were added to pure α-Syn monomer at 10% mass concentration and incubated again at 50 rpm and 37 °C for 72 h. The fibrils formed were centrifuged, resuspended in the desired buffer and the concentration was adjusted accordingly. Similarly, Aβ42 monomer was dissolved in PBS, pH ~7,4 at a concentration of ~25 µM. It was incubated for 24 hours at 37 °C at a rotation of ~350 rpm. The final concentration was adjusted by centrifuging and resuspending in the desired buffer in the required volume. Buffer condition and final protein concentration are mentioned in details in their respective experimental sections.

### Kinetics of Aβ42 aggregation

Lyophilized MAβ42 (~2 mg, MAβ42 represents extra methionine residue at the very N-terminus position) was incubated with 1 mL of 8 M Gdn-HCl for 30 minutes and subsequently purified on a Superdex 75 increase column (GE Healthcare, Chicago, USA) connected to an AKTA pure protein purification system (GE Healthcare, Chicago, USA) with a flow rate of typically 0.7 ml/min and sodium phosphate buffer (20 mM, pH 8, supplemented with EDTA 0.2 mM) as elution buffer. Buffer exchange was carried out for WT proSP-C BRICHOS and its variants with the same sodium phosphate buffer on Zeba spin desalting columns (Thermo Fisher Scientific, Waltham, MA, USA). The individual samples were prepared as follows: ThT (for 20 µM final concentration), sodium phosphate buffer (20 mM, pH 8, supplemented with EDTA 0.2 mM), proSP-C BRICHOS and variants thereof at concentrations varying between 0 µM and 2 µM and monomeric MAβ42 (2 µM). To follow the aggregation kinetics of MAβ42 in presence WT proSP-C BRICHOS domain and its variants, the fluorescence emission of ThT was recorded over time at $\lambda_{em}$ = 480 nm (excitation: $\lambda_{ex}$ = 440 nm, sample volume 100 µl per well) by incubating the samples in a 96-Well Costar Half-Area Black with Clear Flat Bottom Polystyrene NBS Microplate (Corning, USA). Measurements were recorded in a FLUOstar OPTIMA plate reader (BMG Labtech, Ortenberg, Germany) at 37 °C. The data was analyzed using amylofit[13].

### Kinetics of α-Syn aggregation. Kinetics of α-Syn aggregation

Lyophilized α-Syn (~2 mg) was incubated with 1 ml of 8 M Gdn-HCl for 30 minutes and subsequently purified on a Superdex 75 increase column (GE Healthcare, Chicago, USA) connected to an AKTA pure protein purification system (GE Healthcare, Chicago, USA) with a flow rate of typically 0.7 ml/min and sodium phosphate buffer (20 mM, pH 4.8, supplemented with EDTA 0.2 mM) as elution buffer. Buffer exchange was performed for WT proSP-C BRICHOS and its variants with the same sodium phosphate buffer on Zeba spin desalting columns (Thermo Fisher Scientific, Waltham, MA, USA). The individual samples were prepared and concentration was adjusted as follows: ThT (20 µM final concentration), sodium phosphate buffer (20 mM, pH 4.8, supplemented with EDTA 0.2 mM), proSP-C BRICHOS and variants thereof at concentrations varying between 0 µM and 10 µM and monomeric α-Syn (20 µM). To follow the aggregation kinetics of α-Syn in the presence of proSP-C BRICHOS, the fluorescence emission of ThT was monitored over time at $\lambda_{em}$ = 480 nm (excitation: $\lambda_{ex}$ = 440 nm, samples volume 100 µl per well). Protein samples were incubated in 96-Well Costar Half-Area Black with Clear Flat Bottom Polystyrene NBS Microplate (Corning, USA), and measurements were recorded in a

FLUOstar OPTIMA plate reader (BMG Labtech, Ortenberg, Germany) at 37 °C. The data was analyzed using amylofit[13].

## Aggregation kinetics of C-terminal truncated α-Syn (1-121) in presence of proSP-C BRICHOS Var II

Low molecular weight C-terminal truncated α-Syn (1-121) was prepared in PBS buffer by passing through 100 kDa cut-off filter and the flow through was collected. The concentration was adjusted to 150 μM. C-terminal truncated α-Syn (1-121) was incubated at 37 ° with slight agitation (~50 rpm) in the presence and absence of different molar ratio of proSP-C BRICHOS Var II, (i.e. molar ratio of α-Syn: BRICHOS corresponds to 1:0.25, 1:0.5 and 1:1). During aggregation kinetics at regular interval, 10 μl protein solution was taken, diluted to 200 μl PBS and 10 μM ThT was added to the solution. ThT fluorescence was monitored by exciting at 450 nM, while the emission spectrum was recorded in the range of 465-500 nM with slit width 5/5 using Fluoromax-4 Horiba spectrofluoremeter. The Maximum ThT signal i.e., at 482 nM was extracted and was plotted against different time points.

## Microfluidic Diffusional Sizing

Microfluidic diffusional sizing experiments were conducted as described elsewhere. Fabrication and operation of the microfluidic devices was performed as previously described[58,74,75]. The devices were fabricated from PDMS using standard soft-lithography techniques and, subsequently, the surface of both the device and a glass coverslip were activated using oxygen plasma. Sample and buffer were loaded onto the chip from reservoirs connected to the sample and buffer inlets, respectively, by applying a negative pressure at the outlet with a glass syringe (Hamilton, Bonaduz, Switzerland) connected to a syringe pump (neMESYS, Cetoni GmbH, Korbussen, Germany). Imaging was performed using a custom-built inverted epifluorescence microscope equipped with a charge-coupled-device (CCD) camera (Prime 95B, Photometrics, Tucson, AZ, USA) and bright-field LED light sources (Thorlabs, Newton, NJ, USA), using the Cy5-4040C-000 Filter set from Semrock (Laser 2000, Huntingdon, UK) for detection of Alexa647-labelld BRICHOS. Images were typically taken at a flow rate of 100 μl/h, and lateral diffusion profiles were recorded at 4 different positions along the microfluidic channels. From these images, diffusion profiles were extracted using a custom-written analysis code by numerical model simulations by solving the diffusion-advection Eq. 1 for mass transport under flow [76]. This allows determining the fraction of chaperone that is not bound to the fibrillar species. The fraction bound ($f_b$) can be related to the concentration of Brichos ([B]), the concentration of the amyloid fibril ([F]), the dissociation constant ($K_d$) and the stoichoiemtric ratio of Brichos to monomer equivalents in the fibril as shown in Eq. 1.

$$f_b = \left( \frac{[F] + n[B] + K_d - \sqrt{([F] + n[B] + K_d)^2 - 4F[B]}}{2} \right) \frac{1}{n[B]} \qquad (1)$$

The plateau value corresponds to the maximal fraction bound. The concentration of one of the fibrils was varied between 0.01 μM and 24 μM accordingly, while the concentration of labeled BRICHOS was held constant per curve. As we are fitting two unknown parameters simultaneously (i.e. stoichiometry and affinity), three curves at three different concentrations of BRICHOS were measured.

## NMR spectroscopy for structure determination

The NMR experiments were performed on 600 and 700 MHz Bruker spectrometers equipped with a triple resonance cryoprobe at either 298 K or 303 K. Processing and analysis of the NMR spectra were done using NMRPipe, Topspin3.6, CcpNMR, Sparky, and XEASY. The backbone assignments of the WT proSP-C -BRICHOS trimer were performed after recording a set of NMR experiments allowing to connect and identify the residues in a known sequence: HNCA with $40(t_{1,max}(^{15}N) = 9 ms)*128(t_{2,max}(^{13}C) = 12 ms)*2048(t_{3,max}(^1H) = 82 ms)$ complex points; HN(CO)CA with $112(t_{1,max}(^{13}C) = 11 ms)*104(t_{2,max}(^{15}N) = 24 ms)*2048(t_{3,max}(^1H) = 60 ms)$ complex points; HNCO with $140(t_{1,max}(^{13}C) = 23 ms)*44(t_{2,max}(^{15}N) = 10 ms)*2048(t_{3,max}(^1H) = 114 ms)$ complex points; HN(CA)CO with $128(t_{1,max}(^{13}C) = 21 ms)*44(t_{2,max}(^{15}N) = 12 ms)*2048(t_{3,max}(^1H) = 136 ms)$ complex points; and HNCaCb with $96(t_{1,max}(^{15}N) = 20 ms)*128(t_{2,max}(^{13}C) = 5 ms)*2048(t_{3,max}(^1H) = 116 ms)$ complex points. For all the experiments the interscan delay was 0.6 s and the number of scans per increments was 16, except for the HNCACB with 32 scans per increment. For each spectra the data were zero-filled up to 2048 points. The side chains assignments were obtained combining the data from the HNCACB and the HCCH-TOCSY recorded with $200(t_{1,max}(^1H) = 11 ms)*120(t_{2,max}(^{13}C) = 5 ms)*2048(t_{3,max}(^1H) = 116 ms)$ complex points, an interscan delay of 1.5 s and 8 scans per increment. The NOE distance restraints were recorded with [$^{15}$N,$^{13}$C]−resolved [$^1$H,$^1$H]−NOESY with $400(t_{1,max}(^1H) = 23 ms)*80(t_{2,max}(^{13}C) = 10 ms)*2048(t_{3,max}(^1H) = 116 ms)$ complex points. The $F_2$ dimension was recorded so that it was aliased, an interscan delay of 0.8 s and 8 scans per increments were used, the mixing time was set to 55 ms. All the spectra were zero-filled to 2048 points in the direct ($t_3(^1H)$) dimension, 128 in the $^{15}$N dimension ($t_1$ or $t_2$), and 256 in the $^{13}$C dimension ($t_1$ or $t_2$).

For the proSP-C BRICHOS variant II backbone, side chain, and NOE cross-resonances were assigned using the [$^{15}$N,$^{13}$C]−resolved [$^1$H,$^1$H]−NOESY with $352(t_{1,max}(^1H) = 18 ms)*176(t_{2,max}(^{13}C/^{15}N) = 7.5 ms)*2048(t_{3,max}(^1H) = 106 ms)$ complex points, an interscan delay of 0.8 s and 4 scans per increment. The NOESY spectra were recorded with different mixing times: 20, 40, 55, 70 ms. The sidechain assignments were obtained with a HCCH-TOCSY recorded with $128(t_{1,max}(^1H) = 7 ms)*80(t_{2,max}(^{13}C) = 3 ms)*2048(t_{3,max}(^1H) = 106 ms)$ complex points, an interscan delay of 1.5 s and 16 scans per increment. The backbone assignments were obtained using an HNCA recorded with $40(t_{1,max}(^{15}N) = 9 ms)*128(t_{2,max}(^{13}C) = 12 ms)*2048(t_{3,max}(^1H) = 82 ms)$ complex points, and an HNCACB recorded with $96(t_{1,max}(^{13}C) = 16 ms)*64(t_{2,max}(^{15}N) = 17 ms)*2048(t_{3,max}(^1H) = 122 ms)$ complex points, with 16 and 32 scans, respectively. Both experiments were recorded with 0.8 interscan delay.

## Two-dimensional Nuclear magnetic resonance (NMR) spectroscopy

The [$^{15}$N,$^1$H]- heteronuclear multiple quantum correlation (HMQC) spectra of $^{15}$N-labeled α-Syn was recorded on a Bruker either 700 or 600 MHz Avance III HD spectrometer equipped with a cryogenic probe. The number of data points were 128 or 256 in the indirect dimension for each experiment. [$^{15}$N,$^1$H]-HMQC experiments was performed in desired NMR buffer (mentioned in details in their respective section) containing 3% $D_2O$. The temperature was set to either 298 K or 303 K during the course of the measurement time. All NMR spectra were processed with TopSpin 3.2 (Bruker) and analyzed with Sparky and/ CCPN.

## NMR structure determination

Structure calculations were performed with CYANA[77] (Table S1). NOEs were extracted from the [$^{15}$N,$^{13}$C]−resolved [$^1$H,$^1$H]−NOESY spectra to obtain upper distance bounds. Torsion angle restraints were obtained from chemical shifts with TALOS-N[78,79]. Disulfide bridges for the cysteine pairs 120−148 and 121−189 were restrained with distance restraints. For the trimer, in addition, dihedral angle difference restraints and restraints to minimize differences between symmetry-related distances were applied to maintain the symmetry[20]. A total of 200 conformers were calculated using 30'000 (monomer) or 40'000 (trimer) torsion angle dynamics steps, and the 20 conformers with the lowest CYANA target function values were selected to represent the solution structures of the Brichos monomer and trimer.

## α-Syn fibrils-monomer binding experiment

[15N]-labeled α-Syn monomer was prepared (isolated by size exclusion chromatography) in PBS, pH ~7.4. Buffer was exchanged against 25 mM phosphate buffer, 25 mM NaCl, pH ~6.8 with a PD column. Unlabeled α-Syn fibrils that were prepared in PBS, pH ~7.4, were centrifuged and resuspended in 25 mM phosphate buffer with 25 mM NaCl. 100 μM monomeric [15N]-labeled monomeric α-Syn was incubated with 540 μM α-Syn fibrils (as described earlier[21]) at 4 °C for 2 hours. The [$^{15}$N,$^1$H]-HMQC spectra of [15N]-labeled monomeric α-Syn were recorded on a Bruker 600 MHz Avance III HD spectrometer. NMR signal intensity ratios ($I/I_0$) were determined for each residue by extracting the maximal signal height of the cross-peaks from the respective 2D [$^{15}$N-$^1$H] NMR spectra. Thereafter, unlabelled proSP-C BRICHOS variant II was added at a final concentration of ~100 μM. [$^{15}$N,$^1$H]-HMQC spectra of [15N]-labeled α-Syn monomer in presence of fibrils and BRICHOS were recorded time-resolved and analysed correspondingly. The [$^{15}$N,$^1$H]-HMQC spectrum of [15N]-labeled 100 μM α-Syn monomer alone was also measured for normalization.

## α-Syn fibrils - proSP-C BRICHOS variant II binding experiment by NMR

100 μM [15N]-labeled proSP-C BRICHOS variant II was incubated in the absence and presence of ~600 μM WT α-Syn fibrils and C-terminal truncated α-Syn(1-121) fibrils at 4 ° for 2 hours. The corresponding NMR buffer was 25 mM phosphate, 25 mM NaCl, pH ~6.8. [$^{15}$N,$^1$H]-HMQC spectra of [15N]-labeled proSP-C BRICHOS variant II were recorded on Bruker 700 MHz Avance III HD spectrometer with a cryogenic probe.

## proSP-C BRICHOS Var II - client peptide interaction

[15N]-labeled proSP-C BRICHOS Vart II was prepared in 25 mM phosphate buffer, 25 mM NaCl, pH ~6.8. Two substrate/client peptides KKVVVVVKK (peptide A) and VLEMGSGSGSKKVVVVVKK (peptide B), designed to be able to form a beta hair pin (commercially purchased from GL Biochem, Shanghai) were dissolved in DMSO at a stock concentration of 20 mM. The two substrate peptides were added to the proSP-C BRICHOS variant II separately in a concentration dependent manner. Peptide concentrations used in the titration are 100 μM, 500 μM, 1 mM and 2 mM, respectively. Therefore, DMSO concentration in the solution mixture becomes 2.5% (v/v). The reference experiment of free [15N]-labeled proSP-C BRICHOS variant II was therefore measured in presence of 2.5% DMSO. [$^{15}$N,$^1$H]-TROSY spectra of the reference, and in presence peptides were measured on a Bruker 700 MHz Avance III HD spectrometer attached with a cryogenic probe at 303 K. Chemical shift perturbations were determined and plotted against each amino acid residues.

## proSP-C BRICHOS variant II and α-Syn labeling for single-molecule fluorescence microscopy

proSP-C BRICHOS variant II at a concentration of 100 μM was rebuffered using a Pierce concentrator column 10 K MWCO into an amine labeling buffer (PBS with 0.1 M NaHCO3, pH 8.5.) The protein was incubated with a 10-fold molar excess of Atto647 succinimidyl ester for 1 h at room temperature with consecutive separation of free dye by size exclusion chromatography (PD-10 Sephadex column; Amersham) and exchanged to PBS. Similarly, α-Syn monomer was labeled with Alexa 555 dye and mixed with unlabeled α-Syn monomer at a ratio of 1:100.

## Sample preparation for microscopy

Glass coverslips were ultrasonicated in ethanol for 15 min before plasma cleaning for 3 minutes. 30 μl of α-Syn was added and incubated for at least 30 min, before washing with PBS. Next, 1 vol % bovine serum albumin in PBS was added and incubated for 1 hour at RT. Atto647

labeled Brichos was added at a final concentration of 1-100 nM directly before single-molecule imaging.

## Single-molecule imaging

The sample was illuminated and imaged using a custom-build setup, in brief, a 637 nm Coherent Laser was used for illumination through appropriate filters for Atto 647, an Apoplan 100 × 1.46 NA objective from Nikon was used to collect photons with an Andor iXon Ultra camera. An EM gain of 250 allowed for single-molecule detection in typically 5000–10,000 frames with variable frame times of 10 – 35 ms.

## Data analysis

The acquired raw data was smoothened, furthermore non maximum suppression, and thresholding was used to determine locations of single fluorophores. Selected regions of interest were fitted by a pixelated Gaussian function and a homogeneous photonic background with a maximum likelihood estimator for Poisson distributed data with a custom-written MATLAB script. Localizations with an uncertainty of >30 nm were discarded. For single particle tracking, the localizations were linked with a search radius of 270 nm (3 pixels) and a gap size of 2 frames. For further analysis, tracks shorter than 10 frames were ignored. Diffusion coefficients were calculated per track, to exclude background and static molecules, tracks with $D < 0.05$ μm$^2$/s were excluded. For the data in (Fig. 7), at least 6 fields of views from 2 experimental days were included. Mean square displacement versus time lag plots for all tracks was calculated allowing checking of 1 dimensional diffusion model was applicable (Fig. 7c, Supplementary Fig. 11).

## Reporting summary

Further information on research design is available in the Nature Portfolio Reporting Summary linked to this article.

## Data availability

All data underlying the manuscript will be made available on request. The NMR structures detailed in this work are available in the PBD under accession codes 8OVI (monomer) and 8OX2 (timer). NMR chemical shift has been deposited in the Biological Resonance Magnetic Bank under accession codes 34811 and 34813. The PDB code of the previously published structure used in this study is 2YAD. Source data is provided with this paper. Source data are provided with this paper.

## Code availability

The custom scripts will be shared by the authors upon request.

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

## Acknowledgements

D.G and R.R acknowledge Sinergia, SNF for the funding; D-CHAB, LPC, ETH Zurich for the instrumental facility. E.K. acknowledges the VW Stiftung Experiment grant 95664 and the German Research Foundation (DFG) (KL 3278/2-1) grant.

## Author contributions

Conceptualization: D.G., R.R.; Methodology: D.G., F.T., D.A., H.K., Y.F., and R.R. (over all experiments) S.M., E.K. (single-particle tracking and super-resolution microscopy), M.S., G.K., E.A., L.L., T.P.K. (ThT kinetics and microfluidic experiment), J.W. developed the kinetic model of secondary nucleation interference by proSP- C BRICHOS. Visualization: D.G., P.G., F.T., D.A., R.R., Supervision: R.R., Writing: D.G., R.R. All authors reviewed, edited and approved.

## Funding

## Competing interests

The authors declare no competing interest.
