## [Peer Review File · Nature Communications]

The inhibitory action of the chaperone BRICHOS against the α -Synuclein secondary nucleation pathwayREVIEWER COMMENTS

Reviewer #1 (Remarks to the Author):

In this manuscript Ghosh et al have used an array of sophisticated biophysical methods to elucidate the mechanism of action of the BRICHOS chaperone in preventing the secondary nucleation of α -Syn fibers. Through a combination of NMR, microfluidics, aggregation kinetics measurements, and super resolution fluorescence microscopy the authors provide a detailed mechanistic picture for the chaperone's binding to the α -Syn C-terminal tail, and for its prevention of fiber formation by blocking the secondary nucleation pathway.

Overall this is a very impressive manuscript that uses very sophisticated methods and elegantly designed experiments to investigate the BRICHOS aggregation-prevention mechanism. In addition, the manuscript provides the first full-length structure for the trimeric and monomeric chaperone units.

I have but a few comments that I think need to be addressed prior to publication -

1) The authors indicate that the monomer is the active unit of the chaperone, yet the affinity of the WT trimeric protein to A β 42 fibrils is an order of magnitude higher than that of the monomeric BRICHOS variant. Can the authors provide an explanation for these results?

2) The NMR CSPs caused by BRICHOS binding to peptide B are very small and the majority of the chaperone's residues are affected by the binding. There are, however, visible decreases in intensities for only a subset of residues in the spectrum. I recommend, in addition to CSPs changes, to also analyze the intensity changes and take them into account when mapping the interaction.

3) The WT trimeric BRICHOS, while defined as the inactive chaperone conformation, still shows high aggregation-prevention activity. Does the trimer disassemble into the active monomers upon binding to clients? If so, why do the NMR experiments show no binding of this trimer to peptides? This part is not clear and needs to be better explained.

4) The NMR experiments following the release of the monomeric ¹⁵N-labeled α -Syn from the fibers are very elegant. I was, however, surprised that the increase in signal was merely 25%. Would it be possible to completely block the binding to the fibers by adding higher concentrations of the

chaperone, therefore achieving a much higher signal increase?

5) In the aggregation prevention assays shown in Figure 4a, it would be helpful to have all the graphs with the same x axis.

6) The binding curves for BRICHOS variant II show that the affinity is concentration dependent. Can the authors explain the basis for such a behavior?

7) It would also be helpful to show that, upon the removal of the C-terminal tail of α -syn, the chaperone can no longer prevent α -syn aggregation.

8) typos - line 302 - “[15N,1H]-HSQC NMR spectra were measured on 100 μ M 15N-labeled monomeric α -Syn added to unlabeled...

Line 305 - “labeled monomeric α -Syn transiently *binds to* amyloid fibrils yielding a significant signal decay...

Reviewer #2 (Remarks to the Author):

This manuscript studies two aspects of the SP-C BRICHOS chaperone, the structure and oligomerization and secondary nucleation inhibition of alpha-synuclein and Abeta42. The problem is interesting and important biologically, and several different methods were utilized, providing interesting insight. In particular, structures of inactive trimer and active monomer were elucidated using NMR studies, showing mechanism of binding to client. The effects of these variants on secondary nucleation inhibition was studied, showing differential inhibition. Particularly interesting is the proposed mechanism of sweeping by BRICHOS along the fibril, which would allow a single BRICHOS to protect a stretch of the fibril from secondary nucleation events. Overall, the results would be interesting to the scientific community. However, there are some issues that need to be addressed.

Authors, please discuss the analysis to extract binding affinities and stoichiometries in more detail. In a related note, the stoichiometry values are confusing, since presumably BRICHOS can bind at any location on the fibril. In that case, the stoichiometry should increase as a function of BRICHOS concentration. This seems to also be borne out by the tracking results. Alternatively, there are specific sites on the fibrils that BRICHOS binds to (which would presumably be revealed as pauses in the SPT trajectories).

The idea that a single BRICHOS molecule could protect the system against a number of alpha-synuclein molecules from secondary nucleation by scanning the fibril is interesting, and the authors are invoking a mechanism where BRICHOS can displace alpha-synuclein molecules from the fibril. I felt that while there was good evidence for scanning from SPT, the evidence for displacement of bound synuclein was somewhat indirect. Can the authors perform a more direct test of this idea, perhaps with labeled alpha-synuclein?

Since the diffusion was slowed at higher concentration due to multiple BRICHOS occupation, presumably the characteristics would be more complex than simple free diffusion, since it is constrained by adjacent BRICHOS molecules. Authors please elaborate on the evidence for aspect (which seems only briefly alluded to in Figure 7 caption). Also, authors please discuss how multiple labeled BRICHOS on the fibril were treated in the data analysis at higher concentration, for example, were collision events observed, etc.

Reviewer #3 (Remarks to the Author):

The search for means to combat protein assembly diseases was immeasurably advanced when it was discovered that a secondary nucleation pathway was operative, accounting for the bulk of the fibril mass thanks to its exponential character. The subsequent demonstration that BRICHOS could obstruct this pathway provided the first significant sign that this knowledge of pathway might be turned into a therapeutic treatment. The present work describes an attempt to turn the use of a BRICHOS domain into a tool for treating synucleinopathies. On the whole there is much important new work here that should be presented to the scientific community. But being worthy of dissemination is not the same as being of keen interest to a wide audience, and I cannot recommend it for acceptance to Nature.

The following 4 issues are central to this conclusion:

1. While the structural work appears sound, and is certainly of interest, it is not key to whether this BRICHOS is a treatment, or even how it would interact specifically with the fibrils, other than general speculation. (That is, there is no detailed, validated docking). Indeed, while the paper is titled "The inhibitory action of the chaperone BRICHOS against the alpha-Synuclein secondary nucleation pathway..." it is not till after 200 lines of text that the issue of inhibition even appears! The structural work and the inhibitory findings are independent reports, whose separation is readily visualized without damage to the intellectual content of either. A solid paper on the BRICHOS structure in JMB or other journal is not an embarrassment.
2. The use of the "hydrodynamic radius" to study BRICHOS attachment to the fibrils is fundamentally flawed. The hydrodynamic radius will depend on the fibril length, and the lengths will have a distribution. The net observed "radius" is thus a sensitive function of the distribution, and

BRICHOS will change that distribution.

3. The analysis in Figs 4 and 5 is problematic. Unlike most ThT fluorescence curves, a number of the curves rise and then make a sharp cut to a linear phase of growth. With the number of fit curves superimposed, this behavior becomes suppressed in the visual display, but can be discerned by careful inspection. First of all, it is disturbing to have some additional process that is not even recognized, much less explained. But given the global fitting strategy of the fitting program used, one does not know to what extent the fits are compromised by the anomalous behavior (i.e. what parameters are sensitive to this type of anomaly). Fig 5 is dominated by these effects. For example, the yellow curves at 40 μ M are fit by a single curve of classic shape when the data shows two dramatically different phases. Should one be fitting the first half and ignoring the second, or vice versa? The choice made here—to just ignore the point and fit it as one, surely is the worst of the choices. At least in Fig 4 one could wave off the sharp cut at the end as having little influence on the nucleation kinetics that fade out after the start, but here it is totally different.

4. The authors propose a provocative and novel model of BRIOCHOS moving along a fiber and sweeping off the attached secondary nuclei. This is very creative! And the data does show the movement along fibrils. But it needs exploration, not just declaration. For example, if the BRICHOS can move along the fibril, it is weakly enough bound to allow such motion from molecule to molecule. Yet it must be bound tighter than the secondary nucleus in order to displace it. This is a necessary part of this proposal yet not at all addressed. Likewise, the question of how long a secondary nucleus stays attached is very pertinent (and if anyone knows this, it is the Cambridge group). Once a fibril forms from the secondary nucleus, “sweeping” it off is irrelevant. As this is crucial to the mechanism it is unforgivable that it is not addressed.

In addition, I would note the following less critical concerns:

5. How do the authors know that the decreased diffusion they see at higher concentrations is not due to aggregation of the “walkers” on the fibril?

6. The NMR description is rather opaque.

7. With 14 authors, I'd expect that someone might have read the final version and at least spotted all the places where text ran together. (Line 42, 296, 302, 305, 306, 370)

In addition several indecipherable bits appear which only made a difficult paper worse:

Line 245 Measuring the hydrodynamic radius as a function of the A β 42 concentration allows determining both the binding stoichiometry and the dissociation constant, K_d, of the interaction requesting near thermal equilibrium conditions

Line 277 The hydrodynamic radius was then determined followed by the dissociation constant and binding stoichiometry were elucidated.

Line 316 In addition, a kinetic experiment was performed that monitored the signal of monomeric α -Syn bound with fibrils ... [what signal?]

Line 369 However due to the repetitive structure of amyloids having every 4.7 Å another [sic] C-terminal tail avidity is present yielding binding affinities in the range of hundreds of nM at a stoichiometry of 1 BRICHOS domain to ~7-100 α -synuclein molecules within the amyloid (Fig. 4 and Fig. 5).

Reviewer #4 (Remarks to the Author):

This manuscript by Ghosh et al presents experimental data on how a lung-surfactant Brichos domain acts as a chaperone to inhibit α -synuclein aggregation. The model is appealing, that the low-affinity chaperone uses transient weak interactions to compete with and sweep off α -synuclein monomers from the fibril surface, thus inhibiting secondary nucleation. However, this model is not convincingly proven by the data:

1) The conclusion critically depends on the relative affinity of Brichos compared to peptide monomers to fibril surfaces. Single particle tracking gave information about how fast Brichos moves on the fibril, while microfluidic diffusional sizing gave the hydrodynamic radius of Brichos-bound fibrils. What is the uncertainty in the Rh? How does this Rh uncertainty translate to uncertainty in the stoichiometry and affinity of Brichos binding to Ab42 fibrils and to α -synuclein fibrils? Affinity and stoichiometry are independent quantities: for example, high affinity binding can coexist with a low stoichiometry. It's unclear how a measurement of Rh can give both parameters. Moreover, the authors did not show that Brichos binds fibril surfaces with higher affinity or avidity than peptide monomers.

2) The authors state that Brichos interaction with a client peptide is weak and transient. Please provide a quantitative analysis for this weak binding.

Submission of the revised manuscript NCOMMS-23-15640-T entitled “The inhibitory action of the chaperone BRICHOS against the α -Synuclein secondary nucleation pathway at near-atomic resolution” by Ghosh et al.

We would like to thank the reviewers for their careful reading and suggestions. Please find below point by point answers of the reviewers.

Reviewer #1 (Remarks to the Author):

Reviewer #1: In this manuscript Ghosh et al have used an array of sophisticated biophysical methods to elucidate the mechanism of action of the BRICHOS chaperone in preventing the secondary nucleation of α -Syn fibers. Through a combination of NMR, microfluidics, aggregation kinetics measurements, and super resolution fluorescence microscopy the authors provide a detailed mechanistic picture for the chaperone’s binding to the α -Syn C-terminal tail, and for its prevention of fiber formation by blocking the secondary nucleation pathway.

Overall, this is a very impressive manuscript that uses very sophisticated methods and elegantly designed experiments to investigate the BRICHOS aggregation-prevention mechanism. In addition, the manuscript provides the first full-length structure for the trimeric and monomeric chaperone units.

Response: we would like to thank the reviewer for the careful reading and appreciation.

Reviewer #1 (1): The authors indicate that the monomer is the active unit of the chaperone, yet the affinity of the WT trimeric protein to A β 42 fibrils is an order of magnitude higher than that of the monomeric BRICHOS variant. Can the authors provide an explanation for these results?

Response: Following the criticism of the reviewer, we re-measured the binding affinities along with microfluidics experiments. The binding affinity of Var II to A β fibrils was found to be ~21 nM compared to the binding affinity ~192 nM for WT proSP-C BRICHOS, consistent with finding that Var II (monomer) is the active unit of the chaperone. We have now replaced the figures in the revised version of the manuscript (**Fig. 4 in the main manuscript**).

Fig 1. Binding curve for the interaction between different proSP-C BRICHOS variants and Ab42 fibrils. Left panel: Binding curve for the interaction between A β 42 fibrils and WT BRICHOS at three different BRICHOS concentrations 300 nM (red), 600 nM (magenta), 900 nM (blue), yielding a dissociation constant, $K_d \sim 191.7$ [6.2; 517.0] nM with a stoichiometry of 1 BRICHOS molecule per ~ 8 [5; 12] monomer units in the fibril. Middle panel: Binding curve for the interaction between A β 42 fibrils and proSP-C BRICHOS variant II at three different BRICHOS concentrations, 150 nM (red), 300 nM BRICHOS (magenta), and 600 nM (blue). This data yields a dissociation constant, $K_d \sim 21.4$ [0.1; 290.4] nM with a stoichiometry of 1 BRICHOS molecule per ~ 6 [2; 9] monomer units in the fibril. Right panel. Comparison of the affinities of the different proSP-C BRICHOS variants against A β fibrils with the rate constant $k_1 k_2$ at 7.5 μ M proSP-C BRICHOS concentration. For proSP-C BRICHOS Var I, no affinity has been detectable

Reviewer #1 (2): The NMR CSPs caused by BRICHOS binding to peptide B are very small and the majority of the chaperone's residues are affected by the binding. There are, however, visible decreases in intensities for only a subset of residues in the spectrum. I recommend, in addition to CSPs changes, to also analyze the intensity changes and take them into account when mapping the interaction.

Response. Thank you for your suggestion. You correctly mentioned that CSPs caused by BRICHOS binding to peptide B are very small. Following the suggestion of the reviewers we have made new titration experiments and quantified the proSP-C BRICHOS Var II interaction with the client peptide by CSP (Fig. 3c). The binding affinity has been estimated to be $\sim 800 \mu$ M. We have now included the result in the revised manuscript (Fig. S6).

Furthermore, we also analyzed the ligand-induced intensity changes and mapped them on the proSP-C BRICHOS surface. While weak intensity loss is widespread, the largest losses show a similar pattern when compared to the CSPs change (Fig. 2a and b in the figure bellow) (Fig. 3c revised manuscript). We have now included the figure in the revised manuscript (Fig. S5 in the revised manuscript).

Fig.2. Mapping the interaction site of peptide B on proSP-C BRICHOS. (a) 100 μ M proSP-C BRICHOS Var II was incubated in presence of 1 mM substrate peptide B (VLEMGS GSGSKKVVVVVKK) at 303 K, pH ~6.8 in phosphate buffer. Intensity of different amino acids of proSP-C BRICHOS Var II was monitored in absence (I_0) and presence (I) of peptide B. NMR signal intensity ratios (I/I_0) were determined for each residue by extracting the maximal signal height of the cross-peaks from the respective 2D [15 N, 1 H] NMR spectra. (b) The interaction site between peptide B and proSP-C BRICHOS Var II was mapped onto the structure of proSP-C BRICHOS Var II by plotting ratios below 0.7. (c) 100 μ M proSP-C BRICHOS Var II was titrated with increasing concentration of peptide B. Red colour represents proSP-C BRICHOS Var II (100 μ M) in absence of any peptide, whereas, green, yellow, cyan and blue represents proSP-C BRICHOS Var II in presence of 200 μ M, 500 μ M, 1 mM, and 2 mM peptide, respectively. Dissociation constant of peptide B to proSP-C BRICHOS Var II was determined to be ~ 800 μ M (inset) suggesting weak interactions between peptide B and proSP-C BRICHOS Var II domain. It is stated that the binding affinity could not be determined with high accuracy due to the limitation of the addition of high concentration of peptide B. The titration reveals a weak binding in the order of ~800 μ M (inset). The value has to be interpreted with care because of the limited concentration range of the titration that was possible.

Reviewer #1 (3): The WT trimeric BRICHOS, while defined as the inactive chaperone conformation, still shows high aggregation-prevention activity. Does the trimer disassemble into the active monomers upon binding to clients? If so, why do the NMR experiments show no binding of this trimer to peptides? This part is not clear and needs to be better explained.

Response: We were also puzzled with this finding. Since trimeric WT proSP-C BRICHOS has been defined as an inactive form reported in previous literature¹⁻³, we hypothesized (as also stated by the reviewer) that upon binding to α -Syn fibrils, some population of the trimers dissociates into monomers and act as a secondary nucleation inhibitor. Challenged by the reviewer's criticism the following rational why disassembly on the fibril surface may appear was put on the table: It could be that WT proSP-C BRICHOS trimer disassembles into monomer because of a difference of the local microenvironment on the fibril surface. In our previous study we showed that the pH in the vicinity of the α -Syn fibrils is significantly lower (ca 1.5 pH units) compared to the bulk solution⁴. We thus studied a potential trimer monomer disassembly at acidic pH monitored by multi-angle light scattering. Indeed, as shown in **Figure S9 in the revised manuscript** at a pH 5.8 WT proSP-C BRICHOS partly disassembles to monomer in contrast to pH 6.8. These findings are described in the revised manuscript.

Fig 3 Lowering pH causes dissociation of trimeric proSP-C BRICHOS into monomeric entity. The elution profile and molar mass determined by multiangle light scattering of WT proSP-C BRICHOS at pH 6.8 (green) demonstrates its trimeric form, while at lower pH 5.8 (blue) it is present both as a trimer and a monomeric entity.

Reviewer #1 (4): The NMR experiments following the release of the monomeric ^{15}N -labeled α -Syn from the fibers are very elegant. I was, however, surprised that the increase in signal was merely 25%. Would it be possible to completely block the binding to the fibers by adding higher concentrations of the chaperone, therefore achieving a much higher signal increase?

Response. Thank you for the input. Following your suggestion, we have now performed the monomer release experiment by adding BRICHOS Var II in a concentration dependent manner. As expected, monomeric α -Syn signal has been found to increase significantly from 25% to ~65% upon increasing the proSP-C BRICHOS Var II concentration as demonstrated in the Figure below, which is now incorporated in **Fig 6c,6d in the main manuscript**.

Fig 4. Monomer release attached to α -Syn fibrils upon addition of proSP-C BRICHOS Var II. Monomeric α -Syn release attached to α -Syn fibrils upon addition of proSP-C BRICHOS Var II. The experimental set up is as follows: In a sample with 540 μM unlabeled α -Syn amyloid fibrils 100 μM ^{15}N -labeled monomeric α -Syn is added and incubated for two hours. Next, 100 μM (1:1) or 300 μM (3:1) or 500 μM (5:1) proSP-C BRICHOS Var II were added to the sample at time point 0 and the intensity of the ^{15}N -labeled monomeric α -Syn is measured time-resolved by a ^{15}N -filtered NMR experiment (i.e. ^{15}N , ^1H]-HMQC) yielding after ca 120 min ~ 25% (for 1:1), ~50% (for 1:3) and ~65% (1:5) monomer bound to fibrils were released as plotted in bar diagram

Reviewer #1 (5): In the aggregation prevention assays shown in Figure 4a, it would be helpful to

have all the graphs with the same x axis.

Response. Following the suggestion of the reviewer we have now modified the figure in the revised manuscript as suggested:

Fig 5. Secondary nucleation inhibition by different variant of proSP-C BRICHOS. Aggregation of A β 42 in presence of different proSP-C BRICHOS variants. Aggregation of A β 42 in presence of varying molar equivalent of (left panel) WT, Var I (middle panel), and Var II (right panel). The increase in fibrillar mass on the y-axis was measured as an increase in the fluorescence of thioflavin T (ThT). The points represent individual data points. The solid lines represent the fits as obtained from amylo fit as described by Meisl *et al*⁷

Reviewer #1 (6): The binding curves for BRICHOS variant II show that the affinity is concentration dependent. Can the authors explain the basis for such a behavior?

Response. The binding between ligand and fibrils depends on both the affinity (dissociation constant) and the binding stoichiometry. If more ligand is present, it requires a higher concentration of the fibrils in order to bind all ligands present (full saturation). We account for this in the fitting model, as described in the method section.

Reviewer #1 (6): 7) It would also be helpful to show that, upon the removal of the C-terminal tail of α -syn, the chaperone can no longer prevent α -syn aggregation.

Response: Following the excellent suggestion of the reviewer, we performed aggregation kinetics of C-terminal deleted α -Syn (1-121 amino acid residues) in presence and absence of several concentrations of proSP-C BRICHOS variant II (see figure below and also Fig S10). Briefly, 150 μ M C-terminal deleted α -Syn was incubated in presence of 37.5 μ M, 75 μ M and 150 μ M proSP-C BRICHOS. 150 μ M C-terminal deleted α -Syn alone was incubated as a control. It was found that in contrast to WT α -Syn, proSP-C BRICHOS variant II does not have any effect on C-terminal deleted α -Syn aggregation kinetics. We have included the result in the revised manuscript (**Fig. S10**).

Fig. 6. proSP-C BRICHOS Var II does not interfere with the aggregation kinetics of C-terminal truncated α -Syn (1-121). Aggregation of α -Syn(1-121) ($\sim 150 \mu\text{M}$, pH ~ 7.4 , 37°C in PBS) in presence of varying molar equivalents of proSP-C BRICHOS Var II as indicated was monitored by an increase in the fluorescence of thioavin T (ThT). During aggregation, $10 \mu\text{l}$ sample was collected and diluted in $200 \mu\text{l}$ PBS and ThT signal was monitored at regular interval. The data indicate that proSP-C BRICHOS Var II does not influence the C-terminal truncated α -Syn aggregation.

Reviewer #1 (8): typos - line 302 - “[15N,1H]-HSQC NMR spectra were measured on $100\mu\text{M}$ ^{15}N -labeled monomeric α -Syn added to unlabeled...”

Line 305 - “labeled monomeric α -Syn transiently binds to amyloid fibrils yielding a significant signal decay...”

Response. We have now modified the typos in the revised manuscript. We would like to thank for careful reading.

Reviewer #2 (Remarks to the Author):

This manuscript studies two aspects of the SP-C BRICHOS chaperone, the structure and oligomerization and secondary nucleation inhibition of alpha-synuclein and Abeta42. The problem is interesting and important biologically, and several different methods were utilized, providing interesting insight. In particular, structures of inactive trimer and active monomer were elucidated using NMR studies, showing mechanism of binding to client. The effects of these variants on secondary nucleation inhibition were studied, showing differential inhibition. Particularly interesting is the proposed mechanism of sweeping by BRICHOS along the fibril, which would allow a single BRICHOS to protect a stretch of the fibril from secondary nucleation events. Overall, the results would be interesting to the scientific community. However, there are some issues that need to be addressed.

Response. Thank you for carefully reading the manuscript, appreciation, and suggestions. Reviewer #2 (i): Authors, please discuss the analysis to extract binding affinities and stoichiometries in more detail. In a related note, the stoichiometry values are confusing, since presumably BRICHOS can bind at any location on the fibril. In that case, the stoichiometry should increase as a function of BRICHOS concentration. This seems to also be borne out by the tracking results. Alternatively, there are specific sites on the fibrils that BRICHOS binds to (which would presumably be revealed as pauses in the SPT trajectories).

Response. Following the request of the reviewer, the analysis to extract binding affinities and stoichiometry now have been discussed in details in the materials and method section of the revised manuscript as follows:

Binding affinities and stoichiometries are determined using microfluidic technique. Microfluidic devices were fabricated from PDMS using standard soft-lithography techniques and, subsequently, the surface of both the device and a glass cover slip were activated using oxygen plasma. Sample and buffer were loaded onto the chip from reservoirs connected to the sample and buffer inlets, respectively, by applying a negative pressure at the outlet with a glass syringe (Hamilton, Bonaduz, Switzerland) connected to a syringe pump (neMESYS, Cetoni GmbH, Korbussen, Germany). Imaging was performed using a custom-built inverted epifluorescence microscope equipped with a charge-coupled-device (CCD) camera (Prime 95B, Photometrics, Tucson, AZ, USA) and bright field LED light sources (Thorlabs, Newton, NJ, USA), using the Cy5-4040C-000 Filter set from Semrock (Laser 2000, Huntingdon, UK) for detection of Alexa647-labelled BRICHOS. Images were typically taken at a flow rate of 100 $\mu\text{l/h}$, and lateral diffusion profiles were recorded at 4 different positions along the microfluidic channels. From these images, diffusion profiles were extracted using a custom-written analysis code by numerical model simulations by solving the diffusion-advection equation 1 for mass transport under flow. This allows determining the fraction of chaperone that is not bound to the fibrillar species. The fraction bound (f_b) can be related to the concentration of Brichos ([B]), the concentration of the amyloid fibril ([F]), the dissociation constant (K_d) and the stoichiometric ratio of Brichos to monomer equivalents in the fibril can be determined using the following equation

$$f_b = \left(\frac{[F] + n[B] + K_d - \sqrt{([F] + n[B] + K_d)^2 - 4[F][B]}}{2} \right) \frac{1}{n[B]}$$

The concentration of one of the fibrils was varied between 0.01 μM and 24 μM accordingly, while the concentration of labelled BRICHOS was held constant per curve. As we are fitting two unknown parameters simultaneously (i.e. stoichiometry and affinity), three curves at three different concentrations of BRICHOS were measured.

The point raised by the reviewer on the observed sub stoichiometry in binding between proSP-C BRICHOS variants and α -Syn fibrils is not well understood indeed. As stated by the reviewer if there are specific sites on the fibrils that BRICHOS bind the SPT trajectories should reveal pauses, which is not really observed along the fibril (only at the end of fibrils, Fig. S11). We rather favour the following interpretation: BRICHOS binds at every site and diffuses along it covering about ~ 10 α -Syn within a time frame during which no other BRICHOS can bind yielding the observed stoichiometry.

Reviewer #2 (ii): The idea that a single BRICHOS molecule could protect the system against a number of alpha-synuclein molecules from secondary nucleation by scanning the fibril is interesting, and the authors are invoking a mechanism where BRICHOS can displace alpha-synuclein molecules from the fibril. I felt that while there was good evidence for scanning from SPT, the evidence for displacement of bound synuclein was somewhat indirect. Can the authors perform a

more direct test of this idea, perhaps with labelled alpha-synuclein?

Response. Following the suggestion of the reviewer, we tried a displacement of monomeric fluorescence-labelled α -Syn on the fibril using SPT. The attempt was however unsuccessful due to the high abundant monomeric labelled α -Syn in solution. In return, we performed a more detailed, concentration-dependent and in part time-dependent displacement of ^{15}N -labelled α -Syn by proSP-C BRICHOS Var II by NMR (Fig 6 in the main manuscript, also find below). Our NMR studies reveal displacement of monomeric α -Syn attached to the surface of α -Syn fibrils in proSP-C BRICHOS concentration department manner. With increasing proSP-C BRICHOS concentration from 100 μM to 300 μM to 500 μM monomeric α -Syn signal has been found to increase significantly from 25% to ~50% to ~65%, respectively. Therefore, our NMR studies further strengthen the hypothesis of diffusion-dependent sweeping off monomeric α -Syn from the fibrils.

Fig 7. Monomer displacement attached to α -Syn fibrils upon addition of proSP-C BRICHOS Var II. The experimental set up is as follows: In a sample with 540 μM unlabelled α -Syn amyloid fibrils 100 μM ^{15}N -labelled monomeric α -Syn is added and incubated for two hours. Next, 100 μM (1:1) or 300 μM (3:1) or 500 μM (5:1) proSP-C BRICHOS Var II were added to the sample at time point 0 and the intensity of the ^{15}N -labelled monomeric α -Syn is measured by a ^{15}N -filtered NMR experiment (i.e. $[^{15}\text{N},^1\text{H}]$ -HMQC) yielding after ca 120 min ~ 25% (for 1:1), ~50% (for 1:3) and ~65% (1:5) monomer bound to fibrils were released.

Reviewer #2 (iii): Since the diffusion was slowed at higher concentration due to multiple BRICHOS occupation, presumably the characteristics would be more complex than simple free diffusion, since it is constrained by adjacent BRICHOS molecules. Authors please elaborate on the evidence for aspect (which seems only briefly alluded to in Figure 7 caption). Also, authors please discuss how multiple labelled BRICHOS on the fibril were treated in the data analysis at higher concentration, for example, were collision events observed, etc.

Response. Thanks for pointing this out. We have included a supplementary figure to give further details on the single particle tracking data. A maximum concentration of 100 nM of proSP-C BRICHOS Var II was chosen to exclude overlap of single particle tracks overlapping. The photon statistics can be found in Fig S11. For the different concentrations 1 nM – 100 nM there are no significant differences in the distribution of single molecule brightness. This indicates that no collisions or aggregations occurred. However, partial photobleaching at later stages of the experiment might influence this and there is no apparent workaround to completely exclude that already photo bleached proSP-C BRICHOS Var II molecules are blocking parts of the available binding sites. We furthermore show diffusion coefficients in early and late trajectories. Early proSP-C

BRICHOS Var II trajectories were captured immediately after adding proSP-C BRICHOS variant II while late trajectories were 15 min after proSP-C BRICHOS Var II addition to exclude the possibility of slowed diffusion due to binding sites being occupied by multiple proSP-C BRICHOS Var II molecules. We have now discussed the data analysis in detail in the method section as follows:

The data was analyzed as described before. The acquired raw data was smoothened, furthermore non maximum suppression, and thresholding was used to determine locations of single fluorophores. Selected regions of interest were fitted by a pixelated Gaussian function and a homogeneous photonic background with a maximum likelihood estimator for Poisson distributed data with a custom written MATLAB script. Localizations with an uncertainty of >30 nm were discarded. For single particle tracking, the localizations were linked with a search radius of 234 nm (3 pixels) and a gap size of 2 frames. For further analysis, tracks shorter than 10 frames were ignored. Diffusion coefficients were calculated per track, to exclude background and static molecules, tracks with $D < 0.05 \mu\text{m}^2/\text{s}$ were excluded. For the data in Figure 7, at least 6 fields of views from 2 experimental days were included. Mean square displacement versus time lag plots for all tracks was calculated allowing checking of 1 dimensional diffusion model was applicable (**Fig. 7C, Fig. S11**).

Fig 8. Single particle tracking (SPT) of BRICHOS variants along α -Syn fibrils. For further validation of the SPT data, diffusion coefficients were analyzed at different times throughout the experiments, checking if over time proSP-C BRICHOS would fully decorate α -Syn fibrils. In (A) diffusion coefficients are shown for tracks originating from the first 10 min of the experiment and second half from 10 – 20 min after labeled proSP-C BRICHOS Var II addition at different concentrations 1- 100 nM. (B) Photon statistics are plotted, suggesting similar results for different proSP-C BRICHOS Var II concentrations. For SPT analysis, molecules with low photon budget resulting in a localization precision of > 30 nm are excluded. (C) For specificity of binding, low affinity proSP-C BRICHOS Var I and WT proSP-C BRICHOS are compared, therefore similar number of localizations is plotted. Colocalization analysis resulted in higher index (via Pearson correlation) for WT versus proSP-C BRICHOS Var I. (D) Examples of localization density along α -Syn fibrils are plotted for different WT proSP-C BRICHOS concentrations suggesting an inhomogeneous distribution along the main fibril axis at higher concentration.

Reviewer #3 (Remarks to the Author):

The search for means to combat protein assembly diseases was immeasurably advanced when it was discovered that a secondary nucleation pathway was operative, accounting for the bulk of the fibril mass thanks to its exponential character. The subsequent demonstration that BRICHOS could obstruct this pathway provided the first significant sign that this knowledge of pathway might be turned into a therapeutic treatment. The present work describes an attempt to turn the use of a BRICHOS domain into a tool for treating synucleinopathies. On the whole there is much important new work here that should be presented to the scientific community. But being worthy of dissemination is not the same as being of keen interest to a wide audience, and I cannot recommend it for acceptance to Nature.

Response: We would like to indicate that the manuscript was submitted to Nature Communication and not to Nature.

Reviewer #3 (1). While the structural work appears sound, and is certainly of interest, it is not key to whether this BRICHOS is a treatment, or even how it would interact specifically with the fibrils, other than general speculation. (That is, there is no detailed, validated docking). Indeed, while the paper is titled “The inhibitory action of the chaperone BRICHOS against the alpha-Synuclein secondary nucleation pathway...” it is not till after 200 lines of text that the issue of inhibition even appears! The structural work and the inhibitory findings are independent reports, whose separation is readily visualized without damage to the intellectual content of either. A solid paper on the BRICHOS structure in JMB or other journal is not an embarrassment.

Response: It was the attempt of the present manuscript to elaborate on the secondary nucleation inhibition mechanism of the chaperone proSP-C BRICHOS from a structural biology point of view that include structure determination of the inactive trimer of proSP-C BRICHOS, the 3D structure of an active monomer variant, the “sloppy and weak” mechanism of action of proSP-C BRICHOS with a client peptide using a structure-based approach and then apply this information content to the secondary nucleation mechanism. While we agree with the reviewer that the first structure part could fit within the scope of JMB, it is our opinion that only with the initial structural work showing weak chaperone activity of the monomer variant established through structure-based mutagenesis, which was then used for secondary nucleation inhibition, the elucidating experiments on secondary nucleation inhibition were possible.

Reviewer #3 (2): The use of the “hydrodynamic radius” to study BRICHOS attachment to the fibrils is fundamentally flawed. The hydrodynamic radius will depend on the fibril length, and the lengths will have a distribution. The net observed “radius” is thus a sensitive function of the distribution, and BRICHOS will change that distribution.

Response: We did not use the hydrodynamic radius directly as a measure of the affinity, but we fitted the diffusion profiles obtained from microfluidic experiments with respect to two species – one with the radius of free, unbound BRICHOS, and a larger population of surface bound BRICHOS. We can then determine the intensity of both species present, and thereby determine the fraction of BRICHOS that is bound to the fibril. This measure is highly reproducible and accurate, as previously reported in supplementary Figure 5 in (Schneider et al., Nat. Commun. **2021**, *12*, 5999).

Reviewer #3 (3): The analysis in Figs 4 and 5 is problematic. Unlike most ThT fluorescence curves, a number of the curves rise and then make a sharp cut to a linear phase of growth. With the number of fit curves superimposed, this behavior becomes suppressed in the visual display, but can be discerned by careful inspection. First of all, it is disturbing to have some additional process that is not even recognized, much less explained. But given the global fitting strategy of the fitting program used, one does not know to what extent the fits are compromised by the anomalous behavior (i.e. what parameters are sensitive to this type of anomaly). Fig 5 is dominated by these effects. For example, the yellow curves at 40 μ M are fit by a single curve of classic shape when the data shows two dramatically different phases. Should one be fitting the first half and ignoring the second, or vice versa? The choice made here—to just ignore the point and fit it as one, surely is the worst of the choices. At least in Fig 4 one could wave off the sharp cut at the end as having little influence on the nucleation kinetics that fade out after the start, but here it is totally different.

Response: We have now remeasured all the aggregation kinetics and refitted them globally using previously published models⁷⁻⁹ as shown below

Fig. 9: Monomeric proSP-C BRICHOS Var II inhibits A β 42 fibrillation more efficiently than WT Aggregation of A β 42 in presence of different proSP-C BRICHOS variants. Aggregation of A β 42 in presence of varying molar equivalent of (left panel) WT, Var I (middle panel), and Var II (right panel). The increase in fibrillar mass on the y-axis was measured as an increase in the fluorescence of thioflavin T (ThT). The points represent individual data points. The solid lines represent the fits as obtained from amylo fit as described by Meisl *et al*⁷. Var II has the strongest effect.

Fig. 10: Monomeric proSP-C BRICHOS Var II inhibits α -Syn fibrillation more efficiently than WT Aggregation of α -Syn in presence of different BRICHOS variants. Aggregation of α -Syn in presence of varying molar equivalents of WT (left panel), variant I (middle panel) and variant II (right panel). The increase in fibrillar mass was measured as an increase in the fluorescence of thioflavin T (ThT). The points represent individual data points, the solid lines represent the fits as obtained from amylo fit (Meisl *et al*⁷). Var II has the strongest effect.

Reviewer #3 (4): The authors propose a provocative and novel model of BRIOCHOS moving along a fiber and sweeping off the attached secondary nuclei. This is very creative! And the data does show the movement along fibrils. But it needs exploration, not just declaration. For example, if the BRICHOS can move along the fibril, it is weakly enough bound to allow such motion from

molecule to molecule. Yet it must be bound tighter than the secondary nucleus in order to displace it. This is a necessary part of this proposal yet not at all addressed. Likewise, the question of how long a secondary nucleus stays attached is very pertinent (and if anyone knows this, it is the Cambridge group). Once a fibril forms from the secondary nucleus, “sweeping” it off is irrelevant. As this is crucial to the mechanism it is unforgivable that it is not addressed.

Response: The points raised by the reviewer are answered by several arguments based in part by additional experimental data:

(i) proSP-C BRICHOS variant II interacts with the C-terminal flexible tail of the fibrils as demonstrated by the absence of aggregation interference of a C-terminally truncated α -Syn(1-121) (Fig. S10) and the competition between α -Syn monomer with proSP-C BRICHOS variant II and previously published data that α -Syn monomer interact with the C-terminal flexible tail (Kumari et al.⁴).

Fig 11. . proSP-C BRICHOS Var II does not interfere with the aggregation kinetics of C-terminal truncated α -Syn (1-121). Aggregation of α -Syn(1-121) ($\sim 150 \mu\text{M}$, pH ~ 7.4 , 37°C in PBS) in presence of varying molar equivalents of proSP-C BRICHOS Var II as indicated was monitored by an increase in the fluorescence of thioavin T (ThT). During aggregation, $10 \mu\text{l}$ sample was collected and diluted in $200 \mu\text{l}$ PBS and ThT signal was monitored at regular interval. The data indicate that proSP-C BRICHOS Var II does not influence the C-terminal truncated α -Syn aggregation kinetics.

Fig. 12. Binding experiment of WT proSP-C BRICHOS and its variants to monomeric α -Syn and A β 42. Microfluidic Diffusional Sizing experiment showing no increment of the hydrodynamic radius of proSP-C BRICHOS upon binding with monomeric A β 42 (a) and α -Syn (b),

which confirms the lack of binding events of proSP-C BRICHOS with monomeric A β 42 and α -Syn.

(ii) The competition experiments by solution state NMR measuring the displaced α -Syn (shown in Figure 6) shows that monomeric α -Syn is displaced and thus secondary seed formation is inhibited, which is not only observed, but can also be explained by the different K_d between pro-SP-C BRICHOS binding and monomeric α -Syn binding⁴. Previously, we showed K_d between α -Syn monomer to fibrils is \sim 1 mM, whereas our current study shows the binding affinities of different variants of BRICHOS are within the range of high nM to μ M range. For WT dissociation constant was found to be \sim 695.7 [263.1, 1311] nM, whereas for Var II and WT it was calculated to be \sim 450.2 [109.8, 1005] nM and 1.26 [0.09; 2.96] μ M, respectively. We hypothesize that these optimal binding affinities allow BRICHOS molecules to travel freely from molecule to molecule. This notion was also supported by SPT. As correctly stated by the reviewer that these binding affinities are stronger than the binding affinity between α -Syn monomer to fibrils (\sim 1 mM).

Reviewer #3 (5) How do the authors know that the decreased diffusion they see at higher concentrations is not due to aggregation of the “walkers” on the fibril?

Response. We performed another set of single particle tracking analysis. Comparing diffusion coefficients at the onset of the experiments and after 15 min of exposing α -Syn fibrils with 100 nM pro-SP-C BRICHOS Var II with no significant differences in the diffusion coefficients early and late. This suggests, availability of binding sites is not drastically changed throughout the experiments. We furthermore plotted integrated pro-SP-C BRICHOS Var II locations along fibrils, representing all detected locations throughout the time course of the experiment. The obvious higher abundance of bound pro-SP-C BRICHOS Var II along α -Syn fibrils restricts the available binding sites, and results in sub diffusion not to scale linear with concentration. To exclude that we have a dominant fraction of aggregates of pro-SP-C BRICHOS Var II on the fibres, we plotted the photons per single molecule detection as a histogram showing a dominant fraction of monomeric pro-SP-C BRICHOS Var II. However, as the nature of this experiment also contains photobleaching, we cannot fully exclude that already photobleached pro-SP-C BRICHOS Var II molecules block a subset of the available binding sites. A maximum concentration of 100 nM pro-SP-C BRICHOS Var II was chosen to exclude overlap of single particle tracks overlapping. The photon statistics can be found in **Fig. S11** for the different concentrations 1 nM – 100 nM, showing no significant differences in the distribution of single molecule brightness. This indicates that no collisions or aggregations occurred.

6. The NMR description is rather opaque.

Response: We have now discussed NMR in detail in the revised manuscript as follows.

NMR spectroscopy for structure determination. The NMR experiments were performed on 600 and 700 MHz Bruker spectrometers equipped with a triple resonance cryoprobe at either 298

K or 303 K. Processing and analysis of the NMR spectra were done using NMRPipe, Topspin3.6, CcpNMR, Sparky, and XEASY. The backbone assignments of the WT proSP-C -BRICHOS trimer were performed after recording a set of NMR experiments allowing to connect and identify the residues in a known sequence: HNCA with $40(t_{1,\max}({}^{15}\text{N}) = 9 \text{ ms}) * 128(t_{2,\max}({}^{13}\text{C}) = 12 \text{ ms}) * 2048(t_{3,\max}({}^1\text{H}) = 82 \text{ ms})$ complex points; HN(CO)CA with $112(t_{1,\max}({}^{13}\text{C}) = 11 \text{ ms}) * 104(t_{2,\max}({}^{15}\text{N}) = 24 \text{ ms}) * 2048(t_{3,\max}({}^1\text{H}) = 60 \text{ ms})$ complex points; HNCO with $140(t_{1,\max}({}^{13}\text{C}) = 23 \text{ ms}) * 44(t_{2,\max}({}^{15}\text{N}) = 10 \text{ ms}) * 2048(t_{3,\max}({}^1\text{H}) = 114 \text{ ms})$ complex points; HN(CA)CO with $128(t_{1,\max}({}^{13}\text{C}) = 21 \text{ ms}) * 44(t_{2,\max}({}^{15}\text{N}) = 12 \text{ ms}) * 2048(t_{3,\max}({}^1\text{H}) = 136 \text{ ms})$ complex points; and HNCaCb with $96(t_{1,\max}({}^{15}\text{N}) = 20 \text{ ms}) * 128(t_{2,\max}({}^{13}\text{C}) = 5 \text{ ms}) * 2048(t_{3,\max}({}^1\text{H}) = 116 \text{ ms})$ complex points. For all the experiments the interscan delay was 0.6 s and the number of scans per increments was 16, except for the HNCACB with 32 scans per increment. For each spectra the data were zero-filled up to 2048 points. The side chains assignments were obtained combining the data from the HNCACB and the HCCH-TOCSY recorded with $200(t_{1,\max}({}^1\text{H}) = 11 \text{ ms}) * 120(t_{2,\max}({}^{13}\text{C}) = 5 \text{ ms}) * 2048(t_{3,\max}({}^1\text{H}) = 116 \text{ ms})$ complex points, an interscan delay of 1.5 s and 8 scans per increment. The NOE distance restraints were recorded with $[{}^{15}\text{N}, {}^{13}\text{C}]$ -resolved $[{}^1\text{H}, {}^1\text{H}]$ -NOESY with $400(t_{1,\max}({}^1\text{H}) = 23 \text{ ms}) * 80(t_{2,\max}({}^{13}\text{C}) = 10 \text{ ms}) * 2048(t_{3,\max}({}^1\text{H}) = 116 \text{ ms})$ complex points. The F_2 dimension was recorded so that it was aliased, an interscan delay of 0.8 s and 8 scans per increments were used, the mixing time was set to 55 ms. All the spectra were zero-filled to 2048 points in the direct ($t_3({}^1\text{H})$) dimension, 128 in the ${}^{15}\text{N}$ dimension (t_1 or t_2), and 256 in the ${}^{13}\text{C}$ dimension (t_1 or t_2).

For the proSP-C BRICHOS variant II backbone, side chain, and NOE cross-resonances were assigned using the $[{}^{15}\text{N}, {}^{13}\text{C}]$ -resolved $[{}^1\text{H}, {}^1\text{H}]$ -NOESY with $352(t_{1,\max}({}^1\text{H}) = 18 \text{ ms}) * 176(t_{2,\max}({}^{13}\text{C}/{}^{15}\text{N}) = 7.5 \text{ ms}) * 2048(t_{3,\max}({}^1\text{H}) = 106 \text{ ms})$ complex points, an interscan delay of 0.8 s and 4 scans per increment. The NOESY spectra were recorded with different mixing times: 20, 40, 55, 70 ms. The sidechain assignments were obtained with a HCCH-TOCSY recorded with $128(t_{1,\max}({}^1\text{H}) = 7 \text{ ms}) * 80(t_{2,\max}({}^{13}\text{C}) = 3 \text{ ms}) * 2048(t_{3,\max}({}^1\text{H}) = 106 \text{ ms})$ complex points, an interscan delay of 1.5 s and 16 scans per increment. The backbone assignments were obtained using an HNCA recorded with $40(t_{1,\max}({}^{15}\text{N}) = 9 \text{ ms}) * 128(t_{2,\max}({}^{13}\text{C}) = 12 \text{ ms}) * 2048(t_{3,\max}({}^1\text{H}) = 82 \text{ ms})$ complex points, and an HNCACB recorded with $96(t_{1,\max}({}^{13}\text{C}) = 16 \text{ ms}) * 64(t_{2,\max}({}^{15}\text{N}) = 17 \text{ ms}) * 2048(t_{3,\max}({}^1\text{H}) = 122 \text{ ms})$ complex points, with 16 and 32 scans, respectively. Both experiments were recorded with 0.8 interscan delays.

Two dimensional Nuclear magnetic resonance (NMR) spectroscopy. The $[{}^{15}\text{N}, {}^1\text{H}]$ - heteronuclear multiple quantum correlation (HMQC) spectra of ${}^{15}\text{N}$ -labeled α -Syn was recorded on a Bruker either 700 or 600 MHz Avance III HD spectrometer equipped with a cryogenic probe. The number of data points were 128 or 256 in the indirect dimension for each experiment. $[{}^{15}\text{N}, {}^1\text{H}]$ -HMQC experiments was performed in desired NMR buffer (mentioned in details in their respective section) containing 3% D_2O . The temperature was set to either 298 K or 303K during the course of the measurement time. All NMR spectra were processed with TopSpin 3.2 (Bruker) and analyzed with Sparky and/ CCPN.

NMR structure determination. Structure calculations were performed with CYANA (Table S1). NOEs were extracted from the $[{}^{15}\text{N}, {}^{13}\text{C}]$ -resolved $[{}^1\text{H}, {}^1\text{H}]$ -NOESY spectra to obtain upper

distance bounds. Torsion angle restraints were obtained from chemical shifts with TALOS-N¹⁰. Disulfide bridges for the cysteine pairs 120–148 and 121–189 were restrained with distance restraints. For the trimer, in addition, dihedral angle difference restraints and restraints to minimize differences between symmetry-related distances were applied to maintain the symmetry. A total of 200 conformers were calculated using 30'000 (monomer) or 40'000 (trimer) torsion angle dynamics steps, and the 20 conformers with the lowest CYANA target function values were selected to represent the solution structures of the Brichos monomer and trimer.

α -Syn fibrils-monomer binding experiment. ¹⁵N-labeled α -Syn monomer was prepared (isolated by size exclusion chromatography) in PBS, pH ~7.4. Buffer was exchanged against 25 mM phosphate buffer, 25 mM NaCl, pH ~6.8 with a PD column. Unlabeled α -Syn fibrils that were prepared in PBS, pH ~7.4, were centrifuged and resuspended in 25 mM phosphate buffer with 25 mM NaCl. 100 μ M monomeric ¹⁵N-labeled monomeric α -Syn was incubated with 540 μ M α -Syn fibrils (as described earlier⁴) at 4 °C for 2 hours. The [¹⁵N,¹H]-HMQC spectra of ¹⁵N-labeled monomeric α -Syn were recorded on a Bruker 600 MHz Avance III HD spectrometer. NMR signal intensity ratios (I/I₀) were determined for each residue by extracting the maximal signal height of the cross-peaks from the respective 2D [¹⁵N-¹H] NMR spectra. Thereafter, unlabelled proSP-C BRICHOS variant II was added at a final concentration of ~100 μ M. [¹⁵N,¹H]-HMQC spectra of ¹⁵N-labeled α -Syn monomer in presence of fibrils and BRICHOS were recorded time-resolved and analysed correspondingly. The [¹⁵N,¹H]-HMQC spectrum of ¹⁵N-labeled 100 μ M α -Syn monomer alone was also measured for normalization.

α -Syn fibrils - proSP-C BRICHOS variant II binding experiment by NMR. 100 μ M ¹⁵N-labeled proSP-C BRICHOS variant II was incubated in absence and presence of ~600 μ M WT α -Syn fibrils and C-terminal truncated α -Syn(1-121) fibrils at 4 °C for 2 hours. The corresponding NMR buffer was 25 mM phosphate, 25 mM NaCl, pH ~6.8. [¹⁵N,¹H]-HMQC spectra of ¹⁵N-labeled proSP-C BRICHOS variant II were recorded on Bruker 700 MHz Avance III HD spectrometer with a cryogenic probe.

proSP-C BRICHOS variant II - client peptide interaction. ¹⁵N-labeled proSP-C BRICHOS variant II was prepared in 25 mM phosphate buffer, 25 mM NaCl, pH ~6.8. Two substrate/client peptides KKVVVVVVKK and VLEMGSGSGSKKVVVVVVKK, designed to be able to form a beta hair pin (commercially purchased from GL Biochem, Shanghai) were dissolved in DMSO at a stock concentration of 20 mM. The two substrate peptides were added to the proSP-C BRICHOS variant II separately in a concentration dependent manner. Peptide concentrations used in the titration are 100 μ M, 500 μ M, 1 mM and 2 mM, respectively. Therefore, DMSO concentration in the solution mixture becomes 2.5% (v/v). The reference experiment of free ¹⁵N-labeled proSP-C BRICHOS variant II was therefore measured in presence of 2.5% DMSO. [¹⁵N,¹H]-TROSY spectra of the reference, and in presence peptides were measured on a Bruker 700 MHz Avance III HD spectrometer attached with a cryogenic probe at 303 K. Chemical shift perturbations were determined¹¹ and plotted against each amino acid residues.

7. With 14 authors, I'd expect that someone might have read the final version and at least spotted all the places where text ran together. (Line 42, 296, 302, 305, 306, 370)

Response: We would like to thank you for carefully reading. We have now corrected the text passages indicated in the revised version.

In addition several indecipherable bits appear which only made a difficult paper worse:

Line 245 Measuring the hydrodynamic radius as a function of the A β 42 concentration allows determining both the binding stoichiometry and the dissociation constant, K_d, of the interaction requesting near thermal equilibrium conditions

Line 277 The hydrodynamic radius was then determined followed by the dissociation constant and binding stoichiometry were elucidated.

Line 316 In additions; a kinetic experiment was performed that monitored the signal of monomeric α -Syn bound with fibrils ... [what signal?]

Line 369 However due to the repetitive structure of amyloids having every 4.7 Another [sic] C-terminal tail avidity is present yielding binding affinities in the range of hundreds of nM at a stoichiometry of 1 BRICHOS domain to ~7-100 α -synuclein molecules within the amyloid (Fig. 4 and Fig. 5)

Response. Thank you for carefully reading and pointing out the mistakes. We have thoroughly edited the manuscript and changes have incorporated accordingly in the revised manuscript.

Reviewer #4 (Remarks to the Author):

This manuscript by Ghosh et al presents experimental data on how a lung-surfactant Brichos domain acts as a chaperone to inhibit α -synuclein aggregation. The model is appealing, that the low-affinity chaperone uses transient weak interactions to compete with and sweep off α -synuclein monomers from the fibril surface, thus inhibiting secondary nucleation. However, this model is not convincingly proven by the data:

Reviewer #4 (1): The conclusion critically depends on the relative affinity of Brichos compared to peptide monomers to fibril surfaces. Single particle tracking gave information about how fast Brichos moves on the fibril, while microfluidic diffusional sizing gave the hydrodynamic radius of Brichos-bound fibrils. What is the uncertainty in the Rh? How does this Rh uncertainty translate to uncertainty in the stoichiometry and affinity of Brichos binding to Ab42 fibrils and to α -synuclein fibrils? Affinity and stoichiometry are independent quantities: for example, high affinity binding can coexist with a low stoichiometry. It's unclear how a measurement of Rh can give both parameters.

Response: We did not use the hydrodynamic radius directly as a measure of the affinity, but we fitted the diffusion profiles obtained from microfluidic experiments with respect to two species – one with the radius of free, unbound proSP-C BRICHOS, and a larger population of surface bound proSP-C BRICHOS. We can then determine the intensity of both species present, and thus

determine the fraction of BRICHOS that is bound to the fibril. The data is fit to this measure. This measure is highly reproducible and accurate, as previously reported in supplementary Figure 5 in (Schneider et al., Nat. Commun. **2021**, *12*, 5999)⁹. The fraction of bound monomer at a particular concentration of BRICHOS and fibril depends on both the stoichiometry and the dissociation constant. Through measuring the fraction of bound proSP-C BRICHOS at different concentrations of proSP-C BRICHOS and fibrils, we can infer both parameters of stoichiometry and dissociation constant independently.

Reviewer #4 (2): Moreover, the authors did not show that Brichos binds fibril surfaces with higher affinity or avidity than peptide monomers.

Response: We have calculated the binding affinities of different variant of proSP-C BRICHOS to α -Syn fibrils. Binding affinities (K_d) of WT proSP-C BRICHOS, Var I and Var II to α -Syn fibrils have been determined to be ~ 695 nM, ~ 1.3 μ M, and ~ 450 nM, respectively. In our previous study⁴ we showed the binding affinity between α -Syn monomers to α -Syn fibrils is ~ 1 mM. Furthermore, the kinetics of binding of α -Syn monomer to its fibrils is in the fast NMR time regime (micro-to fast milliseconds), while the pro-SP-C BRICHOS binding kinetics to the fibrils is on the slow time scale (slower than seconds). These arguments are now added in the manuscript.

2) The authors state that Brichos interaction with a client peptide is weak and transient. Please provide a quantitative analysis for this weak binding.

Response: Following the suggestion of the reviewer we have made new experiments and quantified the proSP-C BRICHOS Var II interaction with the client peptide (see below). The binding affinity has been estimated to be ~ 800 μ M. It is worth mentioning here that the value has to be interpreted with care because of the limited concentration range of the titration that was possible. At higher concentration it leads to the aggregation and precipitation of the peptide. We have now included the result in the revised manuscript (**Fig. S6**).

Fig 13 Client peptide B interacts weakly with the proSP-C BRICHOS Var II domain. 100 μ M proSP-C BRICHOS Var II was titrated with increasing concentration of peptide B. Red colour represents proSP-C BRICHOS Var II (100 μ M) in absence of any peptide, whereas, green, yellow, cyan and blue represents proSP-C BRICHOS Var II in presence of 200 μ M, 500 μ M, 1 mM, and 2 mM peptide, respectively. Dissociation constant of peptide B to proSP-C BRICHOS Var II was determined to be ~ 800 μ M (inset) suggesting weak interactions between peptide B and proSP-C BRICHOS Var II domain. It is stated that the

binding affinity could not be determined with high accuracy due to the limitation of the addition of high concentration of peptide B.

References

- 1 Biverstål, H. *et al.* Dissociation of a BRICHOS trimer into monomers leads to increased inhibitory effect on A β 42 fibril formation. *Biochimica et biophysica acta* **1854**, 835-843, doi:10.1016/j.bbapap.2015.04.005 (2015).
- 2 Willander, H. *et al.* High-resolution structure of a BRICHOS domain and its implications for anti-amyloid chaperone activity on lung surfactant protein C. *Proceedings of the National Academy of Sciences of the United States of America* **109**, 2325-2329, doi:10.1073/pnas.1114740109 (2012).
- 3 Wang, W. J., Russo, S. J., Mulugeta, S. & Beers, M. F. Biosynthesis of surfactant protein C (SP-C). Sorting of SP-C proprotein involves homomeric association via a signal anchor domain. *The Journal of biological chemistry* **277**, 19929-19937, doi:10.1074/jbc.M201537200 (2002).
- 4 Kumari, P. *et al.* Structural insights into α -synuclein monomer-fibril interactions. *Proceedings of the National Academy of Sciences of the United States of America* **118**, doi:10.1073/pnas.2012171118 (2021).
- 5 Klotzsch, E. *et al.* Superresolution microscopy reveals spatial separation of UCP4 and F0F1-ATP synthase in neuronal mitochondria. *Proceedings of the National Academy of Sciences of the United States of America* **112**, 130-135, doi:10.1073/pnas.1415261112 (2015).
- 6 Anderluh, A. *et al.* Tracking single serotonin transporter molecules at the endoplasmic reticulum and plasma membrane. *Biophysical journal* **106**, L33-35, doi:10.1016/j.bpj.2014.03.019 (2014).
- 7 Meisl, G. *et al.* Molecular mechanisms of protein aggregation from global fitting of kinetic models. *Nature protocols* **11**, 252-272, doi:10.1038/nprot.2016.010 (2016).
- 8 Thacker, D. *et al.* The role of fibril structure and surface hydrophobicity in secondary nucleation of amyloid fibrils. *Proceedings of the National Academy of Sciences of the United States of America* **117**, 25272-25283, doi:10.1073/pnas.2002956117 (2020).
- 9 Schneider, M. M. *et al.* The Hsc70 disaggregation machinery removes monomer units directly from α -synuclein fibril ends. *Nature communications* **12**, 5999, doi:10.1038/s41467-021-25966-w (2021).
- 10 Shen, Y. & Bax, A. Protein backbone and sidechain torsion angles predicted from NMR chemical shifts using artificial neural networks. *Journal of biomolecular NMR* **56**, 227-241, doi:10.1007/s10858-013-9741-y (2013).
- 11 Williamson, M. P. Using chemical shift perturbation to characterise ligand binding. *Progress in nuclear magnetic resonance spectroscopy* **73**, 1-16, doi:10.1016/j.pnmrs.2013.02.001 (2013).

REVIEWER COMMENTS

Reviewer #1 (Remarks to the Author):

The authors have collected and present new data that address, in my view, all of the issues raised before by the reviewers.

Reviewer #2 (Remarks to the Author):

The authors have done a good job of addressing my previous comments. The work will be of substantial interest to the scientific community.

Reviewer #3 (Remarks to the Author):

Before the specifics, two overall observations are in order. (1) This is good work. New and important information is presented, with credible experiments and generally good analysis. It will not embarrass the authors or the journal to have it published to the scientific community. (2) The authors have energetically responded to the reviewers, in many cases redoing entire experiments. This is impressive and exemplary.

I have one further, and critical, scientific remark, and one overall assessment.

On reflection, the “sweeping” idea is far more complex than they describe. An inhibitor on the surface interferes with one single binding site. (incidentally they over-claim that BRICHOS has to be AT the secondary nucleation site. This is incorrect. It only needs to occlude binding to that site, which it could do by being proximate, for example). But how the inhibitor got to the binding site is irrelevant...from an adjacent site by diffusion or from solution. When it moves from site A to site B, site A is now available. The implicit idea that the authors are making is that the secondary nucleus is being incubated somehow, and so one can interrupt the incubation cycle as a means of inhibition. But there is no data to support such a construct, as opposed to a pure stochastic one in which the secondary site will support a random event. In that case, it is irrelevant when the inhibition occurs. (It’s like a slot-machine player being angered that he is thrown off his machine because “it is going to pay out soon after I’ve been playing so long”) To repeat: this is competitive inhibition, and such a process is agnostic as to when the inhibition happens in time, as opposed to that it simply happens at some time. It simply depends on the fact that there are two competitors for the same site. Note that if consistent residence time actually is a variable, it’s a very important development in understanding these processes, but that data is not in this manuscript. But the idea

of “sweeping off” nuclei is not a justified hypothesis.

So that takes me to the assessment. What actually does this manuscript contribute? It is an exploration of the secondary process inhibition on alpha Synuclein. An exploration. The contributions are important. But there is no striking discovery here—no first illustration of inhibition for example, or the like. I personally don’t see this work as profound so as to warrant rapid publication. Others may differ.

As a postscript, the manuscript still shows considerable inattention to detail. For example, ref 17, 18, 27 and 32 have no page numbers. And there is still at least one head-scratching phrase, “...interaction requesting near thermal conditions.”

Reviewer #4 (Remarks to the Author):

The revised manuscript presents various new data to answer the reviewers’ questions, including my questions about stoichiometry and binding affinities. The authors have addressed my questions well. The work contains a wealth of data of different types: structure, dynamics, diffusion, competitive binding, affinity, etc. The model that emerges from these data, that the monomeric chaperon sweeping off monomeric a-Syn from the fibril surface, although not necessarily fully proven, will likely provide good impetus for future studies.

I have one scientific question:

Fig. 4: why do the authors describe variant I as having minimal impact on fibrillization kinetics? At the highest concentration of 2 μ M inhibitor used, the time to reach half of the maximum fluorescence intensity is about 4.2 hrs for variant I. This is only slightly shorter than the corresponding time of ~5 hrs for WT. So although variant I is a weaker inhibitor than WT, it is not dramatically weaker.

Despite the interesting data, this manuscript contains so many imprecise and muddled explanations, and the writing has so many errors, that the paper’s quality suffers disproportionately compared to the data content. This is strange, as the authors responded to reviewer 3 that “We have thoroughly edited the manuscript...”. This is inconsistent with the numerous writing problems. The following are just some of the mistakes that the authors need to fix:

Fig. 6: the authors present new competition data of Brichos binding and aSyn monomer release. In panel (a), “shown along-side overlay spectra of α -Syn monomer in absence (red, middle) and presence (blue, right) of proSP-C BRICHOS Var II ...” There is no overlay in panel (a). Remove this word.

Fig. 6 caption: “Upon addition of proSP-C BRICHOS Var II signal loss is attenuated attributed to a competitive binding between pro-SP-C BRICHOS and monomeric α -Syn ...”

Please rephrase this as “Upon addition of proSP-C BRICHOS Var II, signal loss is attenuated, which can be attributed to...”

For panel (c) “Monomeric α -Syn release attached to α -Syn fibrils upon addition of proSP-C BRICHOS Var II.” This should be rephrased to “Release of α -Syn fibril bound monomeric α -Syn upon addition of ...”

For the data in (c), please denote the ratio of monomer α -Syn and Brichos in a consistent order throughout and make caption consistent with the figure, i.e.. 1 : 5 and not 5 : 1.

Similarly, in Fig. S10, please indicate the order of ratios, whether it is α Syn : Brichos or Brichos : α Syn.

On page 6, the sentence “This indicates that the trimer is an inactive state of proSPC BRICHOS...” is too strong and is inconsistent with the next 1-2 pages of data that provide supporting evidence of the functional oligomeric state of Brichos. Please revise to “This suggests that ...”

In Fig 2: it would be helpful to add the monomer structure in the WT trimer side by side with panel b, to facilitate comparison of the monomeric variant II structure and the WT Brichos structure. Right now, the difference in the loop orientation and the degree of exposure of the hydrophobic surfaces cannot be seen.

On page 9, please put the long “which may have the capacity to form a β -hairpin and shows a more extended hydrophobic segment), representing part of the transmembrane helical segment of native proSP-C,” into a separate sentence.

Page 9, this sentence is muddled and wrong at multiple places. “In contrast to the interaction between the domain and substrate peptide in vivo [missing subject] is likely to be stronger [than what??] as the substrate peptide is covalently linked within [to ??] proSP-C, which furthermore comprises a hydrophobic stretch more than twice as long as our model system.”

Please fix.

Page 9, “It is further noted that more than half of the interaction side lies in the trimer interface” Do you mean “interaction site”?

Page 9, “The interaction between WT proSP-C BRICHOS and A β 42 fibrils showed a significant size increase,”.

Illogical. Interactions do not change size. Rephrase.

Page 9, “determining both the binding stoichiometry and the dissociation constant, K_d , of the interaction requesting near thermal equilibrium conditions”.

The word “requesting” is incorrect. Unclear what the authors mean. Remove or rephrase.

Page 10, “...it is able to inhibit α -Syn aggregation and fibrillation requesting disassembly into its active monomeric state.”.

Do the authors mean “requiring” or “implying” disassembly?

Page 13, “one can urge that slowing down of 1D phenomena could be due to the aggregation of BRICHOS molecules on the α -Syn fibrils. To, exclude this probability, we performed...”

The verb “urge” is out of place. What do you mean?

The comma after “To” should be removed.

Page 13, grammatically incorrect sentence, “Comparing diffusion coefficients at the onset of the experiments and after 15 min of exposing α -Syn fibrils with 100 nM proSP-C BRICHOS Var II with no significant differences in the diffusion coefficients was observed.”

Page 13, another incorrect phrase: “results in sub diffusion not to scale linear with concentration”
At several places the authors describe unfolding of the intrinsically disordered monomeric α -Syn to be aggregated. This seems a contradiction in terms. Aren't intrinsically disordered proteins already unfolded? It seems better to describe the process as folding of the intrinsically disordered monomers to the amyloid fibril structure.

Submission of the re-revised manuscript NCOMMS-23-15640A entitled “The inhibitory action of the chaperone BRICHOS against the α -Synuclein secondary nucleation pathway at near-atomic resolution” by Ghosh et al.

We would like to thank the reviewers for their careful reading and suggestions. We also modified schematic figure 8 as per the suggestion of the editor. Please find below point by point responses to the reviewers’ comments.

Reviewer #1 (Remarks to the Author):

Comment. The authors have collected and present new data that address, in my view, all the issues raised before by the reviewers.

Response. Thank you for the previous suggestions, critical reading, and appreciation of our manuscript.

Reviewer #2 (Remarks to the Author):

Comment. The authors have done a good job of addressing my previous comments. The work will be of substantial interest to the scientific community.

Response. Thank you for your suggestions and careful reading, of our manuscript. We appreciate your interest in our work and your suggestions to improve the manuscript.

Reviewer #3 (Remarks to the Author):

Comment. Before the specifics, two overall observations are in order. (1) This is good work. New and important information is presented, with credible experiments and generally good analysis. It will not embarrass the authors or the journal to have it published to the scientific community. (2) The authors have energetically responded to the reviewers, in many cases redoing entire experiments. This is impressive and exemplary.

Response. Thank you for your suggestions on our previous version of the manuscript, we greatly appreciate your recommendation for the publication of the manuscript.

I have one further, and critical, scientific remark, and one overall assessment.

Comment. On reflection, the “sweeping” idea is far more complex than they describe. An inhibitor on the surface interferes with one single binding site. (Incidentally they over-claim that BRICHOS has to be AT the secondary nucleation site. This is incorrect. It only needs to occlude binding to that site, which it could do by being proximate, for example). But how the inhibitor got to the binding site is irrelevant...from an adjacent site by diffusion or from solution. When it moves from site A to site B, site A is now available. The implicit idea that the authors are making is that the secondary

nucleus is being incubated somehow, and so one can interrupt the incubation cycle as a means of inhibition. But there is no data to support such a construct, as opposed to a pure stochastic one in which the secondary site will support a random event. In that case, it is irrelevant when the inhibition occurs. (It's like a slot-machine player being angered that he is thrown off his machine because "it is going to pay out soon after I've been playing so long") To repeat: this is competitive inhibition, and such a process is agnostic as to when the inhibition happens in time, as opposed to that it simply happens at some time. It simply depends on the fact that there are two competitors for the same site. Note that if consistent residence time actually is a variable, it's a very important development in understanding these processes, but that data is not in this manuscript. But the idea of "sweeping off" nuclei is not a justified hypothesis.

Response. Following the point raised by the reviewer whether there is just a competitive replacement of α -Syn monomer by BRICHOS or also sweeping involved in secondary nucleation formation the following kinetic model present in the Suppl. Material with a summary in the discussion part of the rerevised version of the manuscript is given, showing that sweeping is an important part of the inhibitory mechanism:

Summary in the Discussion:

To strengthen the concept that sweeping of BRICHOS is relevant for secondary nucleation interference, a kinetic model with available quantitative and approximated information was established (see Suppl. Material and **Fig. S12**). The calculations demonstrate that proSP-C BRICHOS interferes with secondary nucleus formation on the surface of the fibrils by diffusing through thousands of potential nucleation sites and thereby kicking α -Syn monomers off the nucleation site via competition. It is the combination of competitive interference with α -Syn monomers and visiting many nucleation sites by diffusion faster than secondary nucleus formation that builds the mechanistic base of its secondary nucleation interference.

Figure 1. Schematic of the kinetic model of proSP-C BRICHOS action in interference with α -Syn secondary nucleation (Fig. S12). M is fibrils mass concentration, m is free monomer concentration, r is the radius of cross section of fibrils, d is the periodic repeat distance of intermolecular β -sheets along the fibril direction, V_m is the volume of one monomer in the fibril, k_2, n_2 are the secondary nucleation rate constant and the secondary nucleation reaction order, D is the

diffusion coefficient of the chaperone on the fibril and k_D is the diffusion limited rate constant of monomers adsorbed on the fibril.

Kinetic Model of Secondary Nucleation Interference by proSP- C BRICHOS

A schematic of the kinetic model of secondary nucleation interference of proSP-C BRICHOS is illustrated in Figure S12. It is based on a cylindrical fibril with radius $r = 5$ nm and α -Syn monomer volume $V_m = 16$ nm³ derived from the cryo EM structure¹. Based on the experimentally derived stoichiometry of 1:10 chaperone molecules (proSP-C BRICHOS: α -Syn monomer) adsorbed α -synuclein molecule in the fibril, we obtain an area $A = 10 \cdot V_m \cdot 2/r = 64$ nm² that each chaperone needs to sweep.

Following the equation $t = AD^{-1}$ it results $t \approx 6.4 \cdot 10^{-4}$ s for proSP-C BRICHOS to move through the area A , where $D = 0.1 \mu\text{m}^2/\text{s}$ is the diffusion coefficient of the chaperone on the fibril.

Next the number of secondary nuclei generated in the area A during the time t is calculated using previously reported literature². First, we consider the total number of secondary nuclei formed per unit time for a system with a (monomer mass equivalent) concentration M of fibrils (10^{-5} mol/L) and a concentration of monomer of m (10^{-5} mol/L):

$$dN = N_A k_2 m^{n_2} M dt = N_A k_2 m^{n_2} N_{\text{agg}} / N_A dt = k_2 m^{n_2} N_{\text{agg}} dt$$

where N_{agg} is the number of monomers per unit volume.

Using the cylindrical geometry of the fibril $N_{\text{agg}} = \pi r^2 h / V_m$ and the number of monomers per area is $r/2V_m$, the number of secondary nuclei formed in time t on a given surface area A is determined:

$$N = k_2 m^{n_2} \frac{r}{2V_m} \cdot A \cdot t$$

where k_2 , n_2 are the secondary nucleation rate constant and the secondary nucleation reaction order, respectively²⁻⁶.

Evaluating for the fibril geometry and $A = 64$ nm², proSP-C BRICHOS diffusion time through the area A $t = 6.4 \cdot 10^{-4}$ s and $k_2 m^{n_2} \approx 1.5 \cdot 10^{-6} \text{s}^{-1}$ yields $N \approx 9.6 \cdot 10^{-9} \ll 1$. This suggests that proSP-C BRICHOS interferes with secondary nucleation formation by passing a (many) nucleation

site(s). With other words, the passing is much faster than the formation of the secondary nucleus. While passing BRICHOS competes with the binding site of α -Syn monomer kicking it off the fibril. Albeit the readsorption of α -Syn monomer is on the time scale of microseconds as determined by NMR experiments ⁸ and by an estimation of a diffusion limited rate constant (see below) and thus many orders of magnitude slower than the secondary nucleation formation. Therefore, by swiping along the fibril surface proSP-C BRICHOS is perturbing the formation of the secondary nucleation.

The readsorption time of α -Syn monomer is given by:

$$t_{ad} = \frac{0.5 \cdot m}{k_D \cdot m \cdot M} = \frac{0.5}{k_D \cdot M}$$

$$D_{mM} = \frac{k_B T}{6\pi\eta} \left(\frac{1}{R_M} + \frac{1}{R_m} \right) \approx D_m = 100 \mu m^2 / s$$

Where $K_D = 4\pi D_{mM} \cdot R_{mM}$ is the diffusion limited rate constant of monomers adsorb on fibrils D_{mM} is coefficient of mutual diffusion of monomers and fibrils and R_{mM} is the encounter distance between monomer and fibril for adsorption happens

Taken together this yields a $t_{ad} = 6.6 \cdot 10^{-5}$ s, which is similar to the experimental value.

Reviewer #4 (Remarks to the Author)

Comment. The revised manuscript presents various new data to answer the reviewers' questions, including my questions about stoichiometry and binding affinities. The authors have addressed my questions well. The work contains a wealth of data of different types: structure, dynamics, diffusion, competitive binding, affinity, etc. The model that emerges from these data, that the monomeric chaperon sweeping off monomeric α -Syn from the fibril surface, although not necessarily fully proven, will likely provide good impetus for future studies.

Response. Thank you for your suggestions, critical reading and appreciation of our manuscript.

I have one scientific question:

Fig. 4: why do the authors describe variant I as having minimal impact on fibrillization kinetics?

At the highest concentration of 2 μ M inhibitor used, the time to reach half of the maximum fluorescence intensity is about 4.2 hrs for variant I. This is only slightly shorter than the corresponding time of \sim 5 hrs for WT. So although variant I is a weaker inhibitor than WT, it is not dramatically weaker.

Response. Thank you for highlighting this point. We agree with the reviewer that proSP-C BRICHOS Var I shows similar inhibitory effect as WT proSP-C BRICHOS against A β 42 fibrillation kinetics. Since the focus is on Var II the statement was overlooked in the revision. We are sorry for this and would like to thank for the careful reading of the reviewer.

Fig. 6: the authors present new competition data of Brichos binding and α Syn monomer release. In panel (a), "shown along-side overlay spectra of α -Syn monomer in absence (red, middle) and

presence (blue, right) of proSP-C BRICHOS Var II ...” There is no overlay in panel (a). Remove this word

Response. We have removed the overlay word in the revised manuscript. The figure caption is now revised as follows.

ProSP-C BRICHOS interacts with the flexible C-terminal part of α -Syn fibrils, which is the secondary nucleation site. **(a)** Competition experiment on α -Syn fibrils (Syn fib) with ^{15}N -labeled α -Syn monomer (Syn Mono) measured by [^{15}N , ^1H]-HMQC experiments against the addition of proSP-C BRICHOS (BRI) Var II. The [^{15}N , ^1H]-HMQC of ^{15}N -labeled α -Syn monomer only (orange, left) is the reference spectrum yielding the I_0 values for panel b, shown along-side [^{15}N , ^1H]-HMQC spectra of α -Syn monomer in absence (red, middle) and presence (blue, right) of proSP-C BRICHOS Var II while bound to α -Syn fibrils.

Fig. 6 caption: “Upon addition of proSP-C BRICHOS Var II signal loss is attenuated attributed to a competitive binding between pro-SP-C BRICHOS and monomeric α -Syn ...” Please rephrase this as “Upon addition of proSP-C BRICHOS Var II, signal loss is attenuated, which can be attributed to...”

Response. Following the suggestion of the reviewer we have now rephrased the sentence as follows “Upon addition of proSP-C BRICHOS Var II, signal loss is attenuated, which can be attributed to a competitive binding between proSP-C BRICHOS and monomeric α -Syn on the fibrils”.

For the data in (c), please denote the ratio of monomer α -Syn and Brichos in a consistent order throughout and make caption consistent with the figure, i.e. 1 : 5 and not 5 : 1. Similarly, in Fig. S10, please indicate the order of ratios, whether it is a Syn : Brichos or Brichos : α Syn.

Response. Following the suggestion of the reviewer, we have incorporated the changes and made the figure captions consistent with the figures in the revised manuscript.

In Fig. S10, the ratios correspond to “ α -Syn : BRICHOS ” . We have now indicated this in the figure caption.

On page 6, the sentence “This indicates that the trimer is an inactive state of proSP-C BRICHOS...” is too strong and is inconsistent with the next 1-2 pages of data that provide supporting evidence of the functional oligomeric state of Brichos. Please revise to “This suggests that ...”

Response. Following the suggestion of the reviewer, we have now revised “indicates that” to “suggests that” in the sentence.

In Fig 2: it would be helpful to add the monomer structure in the WT trimer side by side with panel b, to facilitate comparison of the monomeric variant II structure and the WT Brichos structure. Right now, the difference in the loop orientation and the degree of exposure of the hydrophobic surfaces cannot be seen.

Response. Thank you for your suggestion. In Fig. 2D, the overlay structure of one monomer from the WT proSP-C BRICHOS trimer (green) and the monomer of the mutant proSP-C BRICHOS (Var II) (cyan for β -sheet) has been shown (depicted below). It shows secondary structural elements are conserved in the monomeric proSP-C BRICHOS Var II despite repositioning of the loop.

Figure 2. Structure overlay of one monomer from the WT proSP-C BRICHOS trimer (green) and the monomer of the mutant proSP-C BRICHOS (Var II) (cyan for β -sheet) showing that secondary structure elements are conserved in the monomeric proSP-C BRICHOS Var II despite repositioning of the loop (**Fig. 2D**)

On page 9, please put the long “which may have the capacity to form a β -hairpin and shows a more extended hydrophobic segment), representing part of the transmembrane helical segment of native proSP-C,” into a separate sentence.

Response. Following the suggestion of the reviewer we have now revised the long sentence as follows, “In order to study the chaperone active site of proSP-C BRICHOS, the interaction of the monomeric proSP-C BRICHOS Var II with the client peptides KKVVVVVVVKK (peptide A) and VLEMGS GSGSKKVVVVVVKK (peptide B) was studied by solution state NMR. VLEMGS GSGSKKVVVVVVKK represents part of the transmembrane helical segment of native proSP-C. It may have the capacity to form a β -hairpin and shows a more extended hydrophobic segment”.

Page 9, this sentence is muddled and wrong at multiple places. “In contrast to the interaction between the domain and substrate peptide *in vivo* [missing subject] is likely to be stronger [than what??] as the substrate peptide is covalently linked within [to ??] proSP-C, which furthermore comprises a hydrophobic stretch more than twice as long as our model system.” Please fix.

Response. We have now revised the sentence as follows, “*In vivo* the interaction between the domain of proSP-C BRICHOS and substrate peptide is likely to be stronger compared to our model system (VLEMGS GSGSKKVVVVVVKK i.e. peptide B). The substrate peptide is covalently attached to proSP-C *in vivo* as it is a peptide segment within proSP-C which further comprises of a hydrophobic stretch more than twice as long as our model system”.

Page 9, “It is further noted that more than half of the interaction side lies in the trimer interface” Do you mean “interaction site”?

Response. Thank you for pointing out the typo. We have corrected this in the revised manuscript.

Page 9, “The interaction between WT proSP-C BRICHOS and A β 42 fibrils showed a significant size increase,”
Illogical. Interactions do not change size. Rephrase.

Response. Thank you for your comment. We have now rephrased the sentence to explain our results from the microfluidic diffusional Sizing assay. Hydrodynamic radius of WT proSP-C BRICHOS was monitored before and after interaction with A β 42 fibrils. Increase in hydrodynamic radius of WT proSP-C BRICHOS indicates a binding event between WT proSP-C BRICHOS and A β 42 fibrils.

Page 9, “determining both the binding stoichiometry and the dissociation constant, K_d, of the interaction requesting near thermal equilibrium conditions”.

The word “requesting” is incorrect. Unclear what the authors mean. Remove or rephrase.

Response. Thank you for pointing this mistake. We have now rephrased the sentence in the revised manuscript as “Measuring the hydrodynamic radius as a function of the A β 42 concentration allows determining both the binding stoichiometry and the dissociation constant”

Page 10, “...it is able to inhibit α -Syn aggregation and fibrillation requesting disassembly into its active monomeric state.”.

Do the authors mean “requiring” or “implying” disassembly?

Response. Thank you for suggesting this. The correct sentence should be stated as “...it is able to inhibit α -Syn aggregation and fibrillation implying disassembly into its active monomeric state”.

Page 13, “one can urge that slowing down of 1D phenomena could be due to the aggregation of BRICHOS molecules on the α -Syn fibrils. To, exclude this probability, we performed...”

The verb “urge” is out of place. What do you mean?

The comma after “To” should be removed.

Response. We have now revised the sentence as follows “However, another possibility of slowing down of 1D phenomenon could be due to the aggregation of BRICHOS molecules on the α -Syn fibrils. To exclude this probability, we performed another set of Single Particle tracking analysis (**Fig. S11**)”.

Page 13, grammatically incorrect sentence, “Comparing diffusion coefficients at the onset of the experiments and after 15 min of exposing α -Syn fibrils with 100 nM proSP-C BRICHOS Var II with no significant differences in the diffusion coefficients was observed.”

Response. We have rephrased this sentence in the text as follows, “We compared the diffusion coefficients at the onset of the experiments and after 15 min of exposing α -Syn fibrils with 100 nM fluorescently-labeled proSP-C BRICHOS Var II. However, we did not observe significant difference in the diffusion coefficients”.

Page 13, another incorrect phrase: “results in sub diffusion not to scale linear with concentration.”

Response. We have now rephrased the sentence as follow. The obvious higher abundance of bound proSP-C BRICHOS Var II along α -Syn fibrils restricts the available binding sites. This results in sub diffusive character, which shows non linear behaviour against concentration as would be expected from a 1-D diffusional model.

At several places the authors describe unfolding of the intrinsically disordered monomeric α -Syn to be aggregated. This seems a contradiction in terms. Aren't intrinsically disordered proteins already

unfolded? It seems better to describe the process as folding of the intrinsically disordered monomers to the amyloid fibril structure.

Response. It is true that α -Syn is intrinsically disorder protein. However, α -Syn in the monomeric form is not fully unfolded. Rather, a transient long-range interaction exists between the N-terminal segment and C-terminal region. These intramolecular interactions result in partially collapsed IDP states of α -Syn that deviate from extended polypeptide chain conformations and prevent them to aggregate further. Any driving force such as point mutation, pH changes in the local environment, presence of some co-factors, temperature etc. can perturb the long-range interaction yielding eventually to exposure of the NAC region accelerating thereby the fibrillation process. To clarify this information and following your suggestion we have now revised the statement and describe the process as “misfolding” of the intrinsically disordered monomers to the amyloid fibril structure.

As an editorial request, please improve Figure 8 on resubmission

Response. As per editorial request we modified Figure 8.

Fig 8. Schematic representation of the secondary nucleation inhibition of α -Syn aggregation and fibrillation by the chaperone proSP-C BRICHOS. α -Syn monomers converted into fibrils during its aggregation pathway. Monomeric synuclein binds to the C-terminal flexible tail of the fibrils allowing acceleration of the aggregation process via secondary nucleation. ProSP-C BRICHOS competes with the same binding site on the fibrils, preventing further binding of monomers and cleaning the existing monomers that are already bound to the fibrils.

References.

- 1 Frey, L. *et al.* On the pH-dependence of α -synuclein amyloid polymorphism and the role of secondary nucleation in seed-based amyloid propagation.
- 2 Gaspar, R. *et al.* Secondary nucleation of monomers on fibril surface dominates α -synuclein aggregation and provides autocatalytic amyloid amplification. *Quarterly reviews of biophysics* 50, e6, doi:10.1017/s0033583516000172 (2017).
- 3 Törnquist, M. *et al.* Secondary nucleation in amyloid formation. *Chemical communications (Cambridge, England)* 54, 8667-8684, doi:10.1039/c8cc02204f (2018).
- 4 Hadi Alijanvand, S., Peduzzo, A. & Buell, A. K. Secondary Nucleation and the Conservation of Structural Characteristics of Amyloid Fibril Strains. *Frontiers in molecular biosciences* 8, 669994, doi:10.3389/fmolb.2021.669994 (2021).
- 5 Iljina, M. *et al.* Kinetic model of the aggregation of alpha-synuclein provides insights into prion-like spreading. *Proceedings of the National Academy of Sciences of the United States of America* 113, E1206-1215, doi:10.1073/pnas.1524128113 (2016).
- 6 Catherine, K. X. *et al.* α -Synuclein oligomers form by secondary nucleation. *bioRxiv*, 2023.2005.2028.542651, doi:10.1101/2023.05.28.542651 (2024).
- 7 Buell, A. K. *et al.* Solution conditions determine the relative importance of nucleation and growth processes in α -synuclein aggregation. *Proceedings of the National Academy of Sciences of the United States of America* 111, 7671-7676, doi:10.1073/pnas.1315346111 (2014).
- 8 Kumari, P. *et al.* Structural insights into α -synuclein monomer-fibril interactions. *Proceedings of the National Academy of Sciences of the United States of America* 118, doi:10.1073/pnas.2012171118 (2021).
- 9 Bertoncini, C. W. *et al.* Release of long-range tertiary interactions potentiates aggregation of natively unstructured alpha-synuclein. *Proceedings of the National Academy of Sciences of the United States of America* 102, 1430-1435, doi:10.1073/pnas.0407146102 (2005).

REVIEWERS' COMMENTS

Reviewer #3 (Remarks to the Author):

The authors have presented a reasonable case to justify the sweeping theory they have proposed, and I am satisfied that the manuscript is now acceptable.

I would most strongly advise the authors (in their vast array) to recognize that the justification they gave is that the reaction is diffusion limited, and such reactions are far more readily tested than the hypothesis they have so strongly espoused. I personally remain skeptical that it is the attachment process that controls secondary nucleation, but it certainly might be. But that itself has profound consequences, and is worthy of further exploration.

We would like to thank the reviewers for the careful reading, and suggestions, as well as appreciation of the manuscript. Please find below point by point answers from the reviewers questions.

Reviewer#3

The authors have presented a reasonable case to justify the sweeping theory they have proposed, and I am satisfied that the manuscript is now acceptable.

I would most strongly advise the authors (in their vast array) to recognize that the justification they gave is that the reaction is diffusion limited, and such reactions are far more readily tested than the hypothesis they have so strongly espoused. I personally remain skeptical that it is the attachment process that controls secondary nucleation, but it certainly might be. But that itself has profound consequences and is worthy of further exploration.

Response. Thank you for recommendation to publish in Nature Communications.

As demonstrated in the kinetic model and shown in super-resolution fluorescence microscopy as well as NMR spectroscopy, proSP-C BRICHOS runs along the fibrillar axis diffusion-dependently sweeping off monomeric α -Syn from the fibrils. The observed mechanism explains how a weakly binding chaperone can inhibit the α -Syn secondary nucleation pathway via avidity where a single proSP-C BRICHOS molecule is sufficient against up to ~7-40 α -Syn molecules embedded within the fibrils. However, we agree with the reviewer that further studies are need to establish the fact that attachment process controls secondary nucleation. We have now included this statement in the text.